# Aerosol optical depth determination in the UV using a four-channel precision filter radiometer

Thomas Carlund[1,2], Natalia Kouremeti[1], Stelios Kazadzis[1,3], Julian Gröbner[1]

[1] Physikalisch-Meteorologisches Observatorium Davos/World Radiation Center (PMOD/WRC), Dorfstrasse 33, CH-7260 Davos Dorf, Switzerland

[2] From 1 April 2017 at Swedish Meteorological and Hydrological Institute, SE-60176 Norrköping, Sweden

[3] Institute of Environmental Research and Sustainable Development, National Observatory of Athens, Greece

*Correspondence to*: Thomas Carlund (thomas.carlund@pmodwrc.ch, thomas.carlund@smhi.se)

**Abstract.** The determination of aerosol properties, especially the aerosol optical depth (AOD) in the ultraviolet (UV) wavelength region, is of great importance in order to understand the climatological variability of UV radiation. However, operational retrievals of AOD at the biological most harmful wavelengths in the UVB are currently only made at very few places. This paper reports on the UVPFR (UV precision filter radiometer) sunphotometer, a stable and robust instrument that can be used for AOD retrievals at four UV wavelengths. Instrument characteristics and results of Langley calibrations at a high altitude site were presented. It was shown that due to the relatively wide spectral response functions of the UVPFR, the calibration constants ($V_0$) derived from Langley plot calibrations underestimate the true extra-terrestrial signals. Accordingly, correction factors were introduced. In addition, the instrument's spectral response functions also result in an apparent airmass dependent decrease in ozone optical depth used in the AOD determinations. An adjusted formula for the calculation of AOD, with a correction term dependent on total column ozone amount and ozone air mass, was therefore introduced. Langley calibrations performed 13–14 months apart resulted in sensitivity changes of ≤1.1 %, indicating good instrument stability. Comparison with a high accuracy standard precision filter radiometer, measuring AOD at 368–862 nm wavelengths, showed consistent results. Also, very good agreement was achieved comparing the UVPFR with AOD at UVB wavelengths derived with a Brewer spectrophotometer, which was calibrated against the UVPFR at an earlier date. Mainly due to non-instrumental uncertainties connected with ozone optical depth, the total uncertainty of AOD in the UVB is higher than the ones reported from AOD instruments measuring in UVA and visible ranges. However, the precision can be high among instruments using harmonized algorithms for ozone and Rayleigh optical depth as well as for air mass terms. For four months of comparison measurements with the UVPFR and a Brewer, the root mean squared AOD differences were found <0.01 at all the 306–320 nm Brewer wavelengths.

# 1 Introduction

One of the most important atmospheric processes related with solar UV attenuation is the absorption and scattering of solar radiation by aerosols (IPCC, 2013, Madronich et al., 2015; UNEP, 2010). The effect of aerosols on solar UV radiation is important as it is linked with the impact on UV radiation on human health (Rieder et al., 2008, Cordero et al., 2009)., atmospheric chemistry (e.g. Gerasopoulos et al., 2012) and the biosphere (Diffey, 1991). Especially in heavily polluted areas, analysis of past data series showed, that the decrease of UVB (wavelength range 280-315 nm) radiation due to aerosol attenuation can become larger than the expected increase of UVB radiation due to the declining ozone levels (e.g. Meleti et al., 2009; Zerefos et al., 2012; De Bock et al., 2014). Thus, the determination of aerosol properties, especially the AOD in both the UVA (315-400 nm) and UVB wavelength region, is of great importance in order to understand the climatological variability of UV radiation. However, even though the aerosol attenuation on the solar UVB wavelength range is higher than the one at longer wavelengths, most of the available surface based and satellite AOD measurements are related to the UVA, visible (VIS) and near infrared (NIR) ranges, because they represent the part of the spectrum associated with the higher solar irradiance levels reaching the Earth's surface.

Concerning AOD measurements at the UV range, the largest surface based aerosol sunphotometric network, the Aerosol Robotic Network (AERONET) (Holben et al., 1998)  includes a number of instruments that are able to measure AOD at 340 nm and 380 nm. In addition, the Global Atmospheric Watch precision filter radiometer network (GAW-PFR) provides AOD at 368 nm (Wehrli, 2008). In order to extrapolate the UVA and visible AOD to the UVB the spectral dependence and the aerosol type is needed. This is because the simple Ångström power law includes a wavelength dependence that is related to the different aerosol types, potentially leading to very poor accuracy of AOD in the UVB determined from extrapolation of accurate AOD values in the visible to near infrared range of the spectrum (Li et al., 2012).

Only few instruments such as the UV multifilter radiometer (UVMFR) (Krotkov et al., 2005; Corr et al., 2009; Kazadzis et al., 2016) can be used to provide AOD retrievals in the UVB wavelength range. The Brewer spectrophotometer is an instrument initially designed for providing total column ozone (TCO) measurements based on the use of direct sun irradiance measured at specific wavelengths in the short UVA and in the UVB range (e.g. Kerr et al., 1985). During the past years, several attempts have been presented in the literature, that showed the use of the above mentioned Brewer measurements in order to retrieve AOD in the UVB (e.g. Marenco et al., 1997; Marenco et al., 2002; Cheymol and De Backer, 2003; Cheymol et al., 2006; Gröbner and Meleti, 2004; Kazadzis et al. 2005; Kazadzis et al. 2007; Meleti et al., 2009; De Bock et al., 2010; Kumharn et al., 2012; De Bock et al., 2014). In addition, Arola and Koskela, 2004 have discussed the uncertainties and possible systematic errors linked with the Brewer related direct sun retrieval for AOD.

Recently, the European COST project EUBREWNET (European Brewer network, www.eubrewnet.org/cost1207), for harmonizing European Brewer spectrophotometer measurements, also aims at including an UVB aerosol optical depth product in the common data processing. Over the course of this project the Physikalisch-Meteorologisches Observatorium Davos/World Radiation Center (PMOD/WRC) has been working on a portable and stable instrument to be used for the intercalibration of the various Brewer instruments. As such, the UVPFR instrument built at PMOD/WRC has been used.

Within this study we present the characterization and calibration of the UVPFR instrument as well as validation through field measurements that have been performed at PMOD/WRC.

## 2 Instruments and sites

### 2.1 PFR and UVPFR

The instrument in focus of this study is the UVPFR sunphotometer, which is a modified version of the precision filter radiometer (PFR) designed and built in the late 1990s at PMOD/WRC in Davos, Switzerland. It measures  direct solar irradiance at the four nominal wavelengths 305, 311, 318 and 332 nm at bandwidths of approximately 1.0-1.3 nm at full width half maximum (FWHM). The detectors are operated in a controlled environment and are exposed to solar radiation only during actual measurements. A Peltier thermostat maintains the ion-assisted deposition filters and silicon detectors at a

constant ($\pm$0.1° C) temperature of 20° C over an ambient temperature range from -20° C to +35° C. A shutter opens for only a few seconds during direct sun measurements to keep dose-related degradation of the filters and detectors to a minimum. The vacuum tight sensor head is filled with dry nitrogen gas. In addition to the information given here, a more detailed description can be found in Ingold et al. (2001).

A recent improvement of the instrument was the addition of an UG11 low pass filter at all four channels to remove out-of-

band leakage that had been observed in the original version of the UVPFR.

The spectral response functions of the UVPFR #1001, used in this study, were measured in the laboratory at PMOD/WRC in February 2016, using an EKSPLA NT 200 tuneable laser (www. ekspla.com) as spectral light source. The resulting effective central wavelengths and FWHM are given in Table 1. The spectral response functions have also been convolved (spectral weighting taking into account each filter's spectral response function) with an extra-terrestrial solar spectrum and the results

are given in column 3 of Table 1. The latter are the wavelengths used for calculating the Rayleigh optical depth for the UVPFR #1001. (The differences in Rayleigh optical depth for the two different sets of effective central wavelengths are <0.0007.) The spectral response functions measured in 2016 were also compared with measurements that were performed at the initial stage of the instrument development, in 1999. The difference in effective central wavelengths was $\leq$0.05 nm at all four wavelengths. For the two shortest and therefore most sensitive wavelengths, the difference was only 0.02 nm.

In order to perform direct sun measurements, the UVPFR is mounted on a sun-tracker so that it is continuously pointing to the Sun. The four photometric channels are measured simultaneously by a commercial data logger system (Campbell Scientific CR10X) with 13 bit resolution. Automatic signal ranging within the PFR and logger system is used to increase the dynamic range to 16 bits. The logger clock is frequently updated to be accurate within 1 second. Signal measurements made at full minutes are averages of 10 samples for each channel made over a total duration of 1.25 seconds and can be considered

as instantaneous values.

The full field of view of the instrument is 2.5° and the slope angle is 0.7°. An optical position sensor monitors the solar pointing within a ±0.5° range. Normally, the air pressure at station level is measured with a relative coarse accuracy (±1.0 hPa) barometer (Vaisala PTB101 or Setra Model 278) connected to the UVPFR logger box.

The standard PFR has the same specifications as the UVPFR, except that the PFR measures at the nominal wavelengths 368, 412, 500 and 862 nm with a 5 nm FWHM bandwidth. The PFR, together with an evaluation of different calibration methods, has been described in detail by Wehrli (2000).

## 2.2 Brewer spectrophotometer

The Brewer spectrophotometer (Kerr et al., 1985) is an instrument designed for automated measurements of solar UV irradiance and through them for the retrieval of atmospheric ozone (total column and vertical profile) and sulphur dioxide ($SO_2$). A special version of the instrument (Mk IV) is also able to measure (total column of) nitrogen dioxide ($NO_2$) in the visible range. For the standard TCO measurements direct solar irradiance (direct sun, DS) is measured quasi simultaneous at predefined wavelengths in the UV. The Brewers are also equipped with a global entrance port through which global irradiance spectra are measured.

AOD can be retrieved from the standard DS measurements (e.g. Cheymol and De Backer, 2003) or spectral DS measurements (Kazadzis et al., 2007). In the current study AOD retrievals from the double monochromator Brewer Mk III #163 at the wavelengths 306.3, 310.1, 313.5, 316.8 and 320.0 have been used.

## 2.3 Measurement sites

The UVPFR was calibrated at the Izaña Atmospheric Observatory (IZO) on the island of Tenerife (28.31° N, 16.50° W) at an altitude of 2373 m. At IZO, the Izaña Atmospheric Research Centre (IARC) manages the Regional Brewer Calibration Center – Europe (RBCC-E) and it is the absolute sun calibration facility of PHOTONS and the Red Ibérica de medida Fotométrica de Aerosoles (RIMA) networks. PHOTONS and RIMA are federated networks of the AERONET. In addition, IZO has been recognized as a World Meteorological Organization - Commission for Instruments and Methods of Observation (WMO-CIMO) testbed for aerosol remote sensing instruments including AERONET and GAW-PFR instrumentation.

The home site of the UVPFR is at PMOD/WRC which is located in Davos in the Swiss Alps (46.81° N, 9.84° E) at an altitude of 1590 m. At PMOD/WRC several world references for meteorological radiation measurements are maintained. Among others it hosts the World Optical depth Research and Calibration Centre (WORCC) which maintains the reference triad of PFRs for the global GAW-PFR AOD network.

## 3. Method

### 3.1 Instrument calibration

Calibration of reference sunphotometers with the Langley technique is preferably performed at high altitude stations since it requires low and stable aerosol load (e.g. Shaw 1983). Difficulties with Langley calibration at a low altitude and urban site, when calibration at a high altitude is not possible, have been discussed by Arola and Koskela (2004) and were recently demonstrated by Diémoz et al. (2016). For instruments measuring at wavelengths affected by absorption in ozone, ideally stable total ozone amount is needed during the Langley related period of measurements. These requirements can be relatively frequently fulfilled at IZO.

During May to August 2015 the UVPFR #1001 was operated at the IZO station, with the exception of the time period from the 20 May to 10 June. In September 2016 the next Langley calibration at IZO was performed. In addition to the favourable measurement conditions an advantage of the IZO station is the co-location with other instruments, such as Brewer spectrophotometers and standard PFR sunphotometers. These instruments measure among others total column ozone and AOD in the 368-862 nm range, respectively. These additional variables are highly valuable and help to determine whether measurement conditions during half days (mornings or afternoons) have been suitable for the so called Langley plot calibrations.

The classic Langley method to determine the calibration constant $V_0$ of each wavelength channel, ($V_0$ being equal to the signal that would have been measured at the top of the atmosphere at mean Sun-Earth distance) has been described in many articles on sunphotometry (e.g. Shaw , 1983) and many variations thereof have been published over the last decades. The method is based on the inversion of the so-called Bouguer-Lambert-Beer's law, leading to

$$ln(R^2V) = ln(V_0) - \delta m \tag{1}$$

where the wavelength dependent quantities $ln(V_0)$ and total optical depth $\delta$ can be determined by least-square methods from a number of cloud-free measurements of $V$ taken at different air masses $m$. $R$ is the actual Sun-Earth distance expressed in fraction to 1 AU. The calibration constant $V_0$ used to be found by linear extrapolation to zero air mass of measurements $V$, corrected to mean Sun-Earth distance, and plotted on a logarithmic scale versus air mass. This method is historically called Langley plot calibration (Langley, 1903).

Using a single, common air mass $m$ for all components of the total optical depth can lead to significant errors in $ln(V_0)$ (e.g. Thomason et al. 1983; Forgan, 1988; Russell et al., 1993; Schmid and Wehrli, 1995; Slusser et al. 2000). Two more accurate variants of the Langley extrapolation used here replace $\delta m$ by the individual air mass and optical depth components for Rayleigh scattering (*r*), ozone absorption (*o*) and aerosol extinction (*a*), i.e. $\delta_r m_r + \delta_o m_o + \delta_a m_a$, and solve either of the equations

$$ln(R^2V) + \delta_r m_r + \delta_o m_o = ln(V_0) - \delta_a m_a \qquad \text{or} \qquad (2)$$

$$ln(R^2V) + \delta_r m_r = ln(V_0) - (\delta_o + \delta_a)m_{2ODw} \qquad (3)$$

for $ln(V_0)$ and aerosol optical depth (Eq. 2) or the sum of the two terms ozone and aerosol optical depth $(\delta_o + \delta_a)$ (Eq. 3). The airmass term $m_{2ODw}$ is the ozone and aerosol optical depth weighted sum of $m_o$ and $m_a$, i.e.

$$m_{2ODw} = (\delta_o m_o + \delta_a m_a)/(\delta_o + \delta_a) \qquad (4)$$

The values of ozone ozone optical depth and AOD at IZO used in Eq. (2) and for calculating $m_{2ODw}$ according to Eq. (4) are calculated from total ozone measured by the RBCC-E Brewer spectrophotometer triad (WMO/GAW, 2015) and from the AOD measured by a standard PFR sunphotometer determining AOD at 368, 412, 500 and 862 nm, extrapolated to the actual UV wavelength using the Ångström relation. Langley calibrations based on Eq. 2, sometimes called refined Langley plots (Schmid and Wehrli, 1995), do not require any *a priori* AOD estimate and ozone changes are taken into account if measured

correctly. On the other hand, based on numerical tests, $V_0$ results of an individual Langley event using Eq. 3 were found less sensitive to errors in $\delta_o$. The reason for this should be that when using Eq. (3) the actual value of $(\delta_o + \delta_a)$ is calculated from the linear fit of the Langley plot data and $\delta_o$ is in this case not calculated directly from (uncertain) ozone cross sections and TCO. Values of $\delta_o$, based on TCO measurements by a Brewer, are still used in the weighting of $m_{2ODw}$. But since $\delta_o$ at the three shortest UVPFR wavelengths are about ten times, or more, higher than $\delta_a$ relatively small errors in $\delta_o$ will not have a

large impact on $m_{2ODw}$, and the following determination of $V_0$. For the Langley calibration of the UVPFR at the IZO station, very accurate measurements of both TCO and AOD(368-862nm) were available. As a result, the average $V_0$s at all UVPFR wavelengths differ 0.2 % or less between the two methods.

From the quality of the linear fit of the Langley plot and using TCO and AOD data from the other instruments, the selection of exact airmass range (within 1.2–2.9), and validity of the Langley plot events was mainly based on subjective judging by

the analyst. During the periods when the UVPFR #1001 was at IZO, 27 accepted Langley plot occasions were found in 2015 and 11 were found in 2016. The resulting $V_0$s from these events in 2015 for the method in Eq. (3) are shown in Fig. 1. In addition to the requirement of stable AOD, for the UVPFR it is important to have a stable ozone amount over the site. Or, when very accurate ozone measurements are available, as from the RBCC-E Brewer triad, small ozone changes during the Langley plot periods can be accounted for. From the Brewer measurements the ozone change during each Langley event

was calculated by fitting a linear function to the available TCO measurement data with respect to airmass. The slope of the fit is the change in ozone per unit airmass. The final $V_0$s are then derived from interpolation to zero ozone change as shown in Fig. 1. From this figure it is also evident that the sensitivity to ozone change is low for the 332 nm channel. The sensitivity increases with decreasing wavelength. For the 305 nm channel there is more than 1 % change in $V_0$ per 1 DU change per unit airmass during a Langley event. Similar results were found for the Langley plots in 2016.

In principle, TCO can also be estimated by the UVPFR itself. It is however believed that Brewer spectrophotometers are superior to the UVPFR in TCO determination. At the same time, it is important to remember that the Langley plot calibration of the UV-PFR becomes dependent on the ozone measurements when these are used to correct for ozone changes during Langley events. In case there is a small airmass dependent error in the Brewer (triad) measurements, there will also be an error in the UVPFR $V_0$s.

As is clear from Fig. 1, taking just a single Langley plot event is not enough, if high accuracy accompanied with uncertainty estimation is aimed at. The (experimental) standard deviation of the $V_0$ of the refined method in 2015 is highest for the shortest wavelength (1.28 %) and smallest for the longest wavelength (0.44 %). The standard deviation of the residuals to the linear fit of $V_0$s from Eq. (3) versus ozone change is 0.99 % at the shortest wavelength, and very close to the standard deviation of $V_0$ of the refined method at the other wavelengths. The standard deviations of the $V_0$s in 2016 were slightly lower than for the larger number of Langley plot results in 2015. In addition, the (experimental) standard deviation of the mean $V_0$ for the two periods was 0.25 % and 0.23 %, respectively, at the shortest wavelength.

The final calibration values are shown in Table 2. Over the slightly more than 1-year period between the calibrations at IZO, the decrease in sensitivity was as small as ≤0.2 % at the two shortest wavelength channels. For the 332 nm wavelength the change was -1.0 % and for the apparently least stable channel (318 nm) the change was -1.1 %. With only one channel just exceeding the goal stability of ≤1 % per year, the stability of the UVPFR #1001 is regarded as satisfactory.

Also the PFR-N24 used in this study was calibrated by the refined Langley method in 2015. This was done at the high altitude station at the Mauna Loa Observatory, Hawaii. After this calibration, the PFR-N24 was included as a new member in the WORCC PFR triad operated at PMOD/WRC.

Both Brewer #163 and the UVPFR #1001 participated in the 10[th] RBCC-E campaign 27 May – 4 June 2015 at the INTA (Instituto de Técnica Aerospacial) El Arenosillo station in southern Spain (37.10° N, 6.73° W, 41 m). In addition to the regular calibration of the ozone measurements, Brewer #163 was also absolutely calibrated for AOD determinations versus the UVPFR #1001 during this RBCC-E campaign. Using this calibration, UV AOD has been determined from Brewer #163 during its measurements at PMOD/WRC in Davos. In addition, as part of the regular operations at PMOD/WRC, the sensitivity of Brewer #163 is monitored by taking measurements against reference lamps through the global entrance port. During the period analysed in this study the irradiance sensitivity of the Brewer varied within ±1.2 %, indicating good stability of the measurements taken both through the global and the direct entrance ports.

## 3.2 Corrections due to the finite FWHM of the UVPFR

Due to the large variation with wavelength of ozone absorption in the UV, spectral transmission measurements need to be performed at well-defined and narrow passbands in this wavelength region. The bandwidth of the UVPFR filters, in the order of 1nm, is significantly narrower than for standard VIS-NIR sunphotometers, but about twice as wide as the slit functions of Brewer spectrophotometers. Therefore, the effect of finite bandwidths was investigated for the UVPFR. Effective central wavelengths and full width at half maximum (FWHM) are given in Table 1.

Due to the very strong increase in ozone absorption with decreasing wavelength, and hence its stronger change with airmass at the shorter wavelength side of the filter band passes, this leads to an increase in the effective wavelengths seen by the UVPFR when the airmass increases. This in turn leads to errors in the extrapolation to zero airmass during a Langley calibration. The FWHM effect has been quantified with simple but high resolution modelling with the Bouguer-Lambert-Beer's law.

Using an extra-terrestrial solar spectrum of 0.05 nm resolution with a 0.01 nm increment (Egli, et al. 2013), together with ozone absorption coefficients for 223 K from Molecular Spectroscopy Lab, Institute of Environmental Physics (IUP), University of Bremen (Serdyuchenko, et al., 2011) , direct solar irradiance spectra at the surface were calculated for different airmasses and TCO amounts. The IUP ozone cross sections were chosen by convenience since they matched the 0.01 nm resolution of the used extra-terrestrial solar spectrum. This was not the case for the cross sections by Bass and Paur (1985) which are used in the operational TCO determinations by the Brewers, as well as for the AOD determinations with both the UVPFR and the Brewer, discussed later in this study. It is assumed that the choice of ozone cross sections does not significantly affect the modelled FWHM effects within their estimated uncertainty.

The aerosol extinction was modelled using the Ångström law, $AOD_\lambda=\beta\lambda^{-\alpha}$ (Ångström, 1929), with the parameters $\alpha$=1.3 and $\beta=AOD_{1000nm}$=0.012,. Ångström (1929) suggested that values of $\alpha$ would normally be within 1.0 to 1.5. From this, and many more recent measurements, the conventional value of 1.3 has emerged, see e.g. Gueymard (1998). During the Langley calibrations at IZO, $\alpha$ determined from AOD retrievals with standard PFR sunphotometer was always found to be less <2, with an average value of about 1.5. With parameters $\alpha$=1.3 and $\beta$ =0.012 AOD at 305 nm becomes 0.056 and this value was slightly higher than the mean value during accepted Langley plot events. In the end, no matter if $\alpha$=2 had been used, with the low aerosol optical depth present at IZO the influence of finite bandwidths due to aerosol extinction varying with wavelength was found to be negligible.

In the calculations a station pressure of 770 hPa was used, which is close to the average value at the IZO station during the evaluated Langley plot events. Effective ozone altitude was set to 25 km and 22 km for calculations corresponding to measurements at Izaña and Davos, respectively. These values on ozone altitude were also used for the Langley calibrations at IZO (Sect 2.2) and for the AOD determinations in Davos (Sect. 5). For the relative optical airmass for ozone absorption the algorithm/formula by Komhyr et al. (1989) was used. Rayleigh optical depth, $\delta_r$, was calculated according to Bodhaine et al. (1999) and the relative optical airmass for Rayleigh scattering was calculated according to Kasten and Young (1989). The aerosol relative optical airmass, $m_a$, was estimated by an algorithm for water vapour airmass, $m_w$ (Gueymard, 1995). The vertical distribution of the aerosol particles is generally not known but also in other AOD calculations the aerosol airmass has been approximated by $m_w$, e.g. for the GAW-PFR network (McArthur et al., 2003; Wehrli, 2008). Finally, the calculated irradiance spectra were convolved with the measured spectral response functions of UVPFR #1001.

Results of Langley plots of the simulated UVPFR direct irradiances, $V_{0,\lambda}$(Langley), were then compared to the extra-terrestrial irradiances calculated by convolving the extra-terrestrial spectrum with the UVPFR spectral response functions, $V_{0,\lambda,true}$. The FWHM effect is mainly dependent on ozone amount and airmass range. On average the air mass range was 1.3-

2.8 and average TCO was 290 DU during the Langley plots at Izaña. For these conditions, the $V_0$ correction factors $c_{FWHM}=$ $V_{0,\lambda,true}/V_{0,\lambda}$(Langley) were estimated to $c_{FWHM}=$[1.012 1.003 1.001 1.000] for the UVPFR channels from the shortest to the longest wavelength.These values are smaller but in line with corrections calculated for 2 nm FWHM using a more comprehensive model (Slusser et al., 2000). Accordingly, Langley extrapolation corrections found for the Brewer spectrophotometer (Gröbner and Kerr, 2001) are smaller than for the UVPFR at corresponding wavelengths, mainly due to the smaller FWHM (0.5–0.6 nm) of the Brewer.

Not only the derived $V_0$s are affected by the FWHM effect due to the rapidly changing ozone absorption with wavelength. Even if the correct $V_0$s are used, the calculated UVB AOD will still be incorrect if no further correction is applied. With increasing airmass there is an increase in effective central wavelength for the sunphotometer channels as mentioned above. This results in an apparent decrease in ozone optical depth with increasing airmass. This effect was quantified by calculating the ozone optical depth from the modelled UVPFR direct irradiance signals using the Rayleigh and aerosol optical depth values at their fixed effective central wavelengths. The effect varies slightly with station altitude/pressure. In Fig. 2, results are shown for an approximate pressure level in Davos ($p=$ 840 hPa). The changes in effective ozone optical depth are strongest for the shortest wavelengths. The effect is negligible at the 332 nm wavelength.

The apparent change in ozone optical depth is not a perfect linear function with airmass. With little loss in accuracy, the ozone optical depth correction is still estimated as a linear function of $m_o$ with the lines passing through the origin. The error in the derived AOD using this simplification is according to the calculations performed here ≤0.001 units of AOD at the shortest wavelength and high total ozone amount, and considerably smaller at the other wavelengths and/or lower *TCO*. The resulting ozone optical depth correction factor for 350 DU total column ozone, $c_{o,\lambda,350DU}$, is given in Table 2. The apparent decrease in ozone optical depth gets stronger with increasing total column ozone. The ozone optical depth change for 350 DU is taken as reference. Then the ratio of the ozone optical depth change at other ozone amounts at a specific $m_o$ is very similar for all wavelengths and can be approximated by a quadratic polynomial as

$$f_{o,DU} = c_{o,\lambda}/c_{o,\lambda,350DU} = 6.1443 \cdot 10^{-6} \cdot \text{TCO}^2 + 0.8518 \cdot 10^{-3} \cdot \text{TCO} - 0.0513 \qquad (5)$$

where TCO is the total column ozone amount expressed in Dobson units. In this case, the coefficients in Eq. (5) are derived for a pressure of 840 hPa, corresponding to normal conditions in Davos. The resulting difference in $f_{o,DU}$ is negligible both for conditions at IZO (about 770 hPa) and at sea level with differences in calculated AOD being less than 0.0005. Finally, the apparent decrease on ozone optical depth, $\Delta\delta_{o,\lambda}$, is calculated as

$$\Delta\delta_{o,\lambda} = f_{o,DU}c_{o,\lambda,350DU}m_o \qquad (6)$$

The Langley plot and AOD modelling was also made for a case with zero total column ozone. This showed that the FWHM effects accounted for above are practically entirely caused by the rapidly increasing ozone absorption with decreasing

wavelength. For example, a similar correction of the Rayleigh optical depth as for the ozone optical depth correction in Eq. (6) would, at any of the UVPFR wavelengths, only be about 1/8 of the $\Delta\delta_{o,\lambda}$, at the 332 nm channel.

All in all, at an airmass of 2 and total column ozone amount of 350 DU the effect of the FWHM corrections on derived AOD at 305 nm is about +0.015, while it is only about +0.004 at 311 nm. Both these values are much lower than the total

uncertainty in the UV AOD (see Sect. 4 below) but since the errors due to the finite FWHM are systematic the relatively small corrections are still performed (GUM, 2008).

## 3.3 Calculation of AOD from UVPFR measurements

A more detailed form of the Bouguer-Lambert-Beer law in Eq. (1), valid at a (monochromatic) UVPFR wavelength $\lambda$, is

$$\ln(R^2 V_\lambda) = \ln(V_{0,\lambda}) - m_r \delta_{r,\lambda} - m_o \delta_{o,\lambda} - m_a \delta_{a,\lambda} - m_n \delta_{n,\lambda} - m_s \delta_{s,\lambda} \tag{7}$$

Solving for aerosol optical depth, $\delta_{a,\lambda}$, and neglecting the assumed very small optical depths due to absorption in $NO_2$ ($\delta_{n,\lambda}$) and $SO_2$ ($\delta_{s,\lambda}$) while including the FWHM corrections $c_{FWHM,\delta}$ and $\Delta\delta_{o,\lambda}$ described above, leads to

$$AOD_\lambda = \delta_{a,\lambda} = ln\left(\frac{c_{FWHM,\lambda} V_{0,\lambda}}{R^2 V_\lambda}\right)\Big/ m_a - \frac{m_r}{m_a}\delta_{r,\lambda} - \frac{m_o}{m_a}\left(\delta_{o,\lambda} + f_{o,DU} c_{o,\lambda,350DU} m_o\right) \tag{8}$$

from the measurement of one of the spectral UVPFR output signals $V_\lambda$. The $V_{0,\lambda}$ is the calibration constant at the same wavelength derived from the Langley plot calibrations as described above.

The ozone optical depth, $\delta_{o,\lambda}$, is calculated from ozone absorption coefficients, $k_{o,\lambda}$, and *TCO* amount. To comply with the

Brewer's operational TCO determinations, the ozone absorption coefficients are based on ozone cross section data determined by Bass and Paur (1985). The effective ozone temperature and altitude are also approximated in the same way as for the Brewer operational ozone amount determinations, i.e. by the constant values -45 °C and 22 km, respectively. Values of $k_{o,\lambda}$ for the UVPFR #1001 are given in Table 2.

Indeed, using different datasets on ozone cross sections would result in different AOD values, especially at the shortest

wavelengths. The effect of different cross sections is not further investigated here. In any case the same cross sections should be used for both TCO and AOD determinations.

The ozone amounts taken from a collocated Brewer are calculated with Rayleigh scattering coefficients according to Nicolet (1984), instead of the standard ones used in the operational Brewer program. As an example, for Brewer #163 in Davos the corrected TCO values are 2.7 DU lower than the operational ones. Using Rayleigh scattering coefficients calculated

according to Bodhaine et al. (1999) gives similar results, within 0.1 DU, as with the coefficients according to Nicolet (1984). The other parameters on the right-hand side of Eq. (8) are calculated mainly from position and time and the applied airmass formulas were given in Sect. 2.3 above. As above, Rayleigh optical depth, $\delta_{r,\lambda}$ , is calculated with the Bodhaine et al. (1999)

algorithm. Air pressure, $p$, is required for the calculation of Rayleigh optical depth and $p$ is also measured at the station. Potential absorption by $NO_2$ and $SO_2$ is not included in Eq. (8). The actual amounts of these gases over the measurement site(s) are assumed to be negligibly small. The potential error of this simplification is quantified in the next section.

AOD values calculated by Eq. (8) are only valid for times when there are no clouds in front of the sun. The cloud screening applied in this study is based on the method by Alexandrov et al. (2004) with modifications to fit the UVPFR measurements. The Alexandrov et al. (2004) cloud screening algorithm was developed for optical depth measurements at 870 nm wavelength, and for a sampling interval of 20 seconds. Stability tests were performed with a 15 measurements window, which consequently spanned over five minutes. For the cloud screening, optical depth at the longest UVPFR wavelength (332 nm) was used. Since the UVPFR only takes measurements once every minute only 5 measurements were used for the stability check. Also, the threshold for the inhomogeneity parameter $\varepsilon'$ was increased from $2 \cdot 10^{-4}$ to $4.5 \cdot 10^{-4}$. A further restriction introduced was that the atmospheric transmission at the shortest UVPFR wavelength (305 nm) had to be >0.001. This did not result in a perfect cloud screening of UVPFR data but it was considered good enough for the analyses in this study. Remaining cloud affected data often caused clear outliers in the comparisons with the PFR and Brewer instruments, which then could be removed.

## 3.4 AOD uncertainty

An uncertainty analysis according to GUM (GUM, 2008) has been made for the AOD values retrieved from a UVPFR sunphotometer. Assume we have an arbitrary measurand with its estimated value, $y$, which is not directly measured but determined from $N$ other estimated quantities $x_1, x_2, \ldots, x_N$ through a functional relationship $y=f(x_1, x_2, \ldots, x_N)$. Then the law of propagation of uncertainties for independent variables states that for the combined standard uncertainty of the measurand estimate $y$, $u_c(y)$,

$$u_c{}^2(y) = \sum_{i=1}^{N}\left(\frac{\partial f}{\partial x_i}\right)^2 u^2(x_i) \tag{9}$$

where $u(x_i)$ is the standard uncertainty of each input variable $x_i$ (GUM, 2008). For the $AOD_\lambda=\delta_{a,\lambda}$ calculated according to Eq. (8) this translates to

$$u_c(\delta_{a,\lambda}) = \left\{\left[\frac{1}{m_a}\frac{u(V_{0,\lambda})}{V_{0,\lambda}}\right]^2 + \left[\frac{1}{m_a}\frac{u(R^2)}{R^2}\right]^2 + \left[\frac{1}{m_a}\frac{u(V_\lambda)}{V_\lambda}\right]^2 + \left[\frac{1}{m_a}\frac{u(V_{CS,\lambda})}{V_{CS,\lambda}}\right]^2 + \left[\frac{m_r}{m_a}u(\delta_{r,\lambda})\right]^2 + \left[\frac{m_o}{m_a}u(\delta_{o,\lambda})\right]^2 + \left[\frac{m_n}{m_a}u(\delta_{n,\lambda})\right]^2 + \right.$$

$$\left. + \left[\frac{m_s}{m_a}u(\delta_{s,\lambda})\right]^2 + \left[\delta_{a,\lambda}\frac{u(m_a)}{m_a}\right]^2 + \left[\delta_{r,\lambda}\frac{u(m_r)}{m_a}\right]^2 + \left[\delta_{o,\lambda}\frac{u(m_o)}{m_a}\right]^2 + \left[\delta_{n,\lambda}\frac{u(m_n)}{m_a}\right]^2 + \left[\delta_{s,\lambda}\frac{u(m_s)}{m_a}\right]^2\right\}^{1/2}$$

$$\tag{10}$$

For simplicity, the contribution due to correlated variables has been omitted. The term $V_{CS,\lambda}$ is the contribution to the measured signal due to additional circumsolar radiation seen within the field of view of the UVPFR, further discussed in

Sect 4.4. Similar AOD uncertainty expressions can be found in the literature (e.g. Russell et al., 1993; Carlund et al., 2003;). A slightly different approach was taken by Mitchell and Forgan (2003) where they investigated uncertainty in total optical depth from different sunphotometers measuring at similar wavelengths. When using Eq. (10), uncertainty of $c_{FWHM}$ is included in the $u(V_{0,\lambda})$ uncertainty and uncertainty contributions from $f_{o,DU}$ and $\varDelta\delta_{o,\lambda,350DU}$ are included in the $u(\delta_{o,\lambda})$ term. To get the expanded uncertainty, $U$, the combined standard uncertainty ($u_c(\delta_{o,\lambda})$) is multiplied by a coverage factor, $k$. In this case $k=2$ is chosen to get an approximate level of confidence of 95 %. So

$$U_{95} = k \cdot u_c(\delta_{a,\lambda}) = 2 \cdot u_c(\delta_{a,\lambda}) \tag{11}$$

for a number of effective degrees of freedom of $u_c(\delta_{a,\lambda})$ of significant size (>50), which is here the case.

### 3.4.1 Uncertainty of ozone optical depth

At the shortest UVPFR wavelengths the most dominant source of uncertainty in AOD determinations originates from the uncertainty in ozone optical depth. In the $u(\delta_{o,\lambda})$, contributions from ozone cross section related uncertainty, uncertainty in TCO amount and effective ozone temperature are taken into account. Bass and Paur only report 1% noise during their measurements (Bass and Paur, 1985). Gorshelev et al. (2014) estimate that the total uncertainty in the Bass and Paur cross sections exceeds 2 %. Serdyuchenko et al. (2011) state that a 3 % accuracy has been achieved for their (IUP Bremen) ozone cross sections and Gorshelev et al. (2014) state 2-3 % total uncertainty for the wavelength region under consideration here. Recently, Weber, et al. (2016) reviewed the uncertainty of ozone cross datasets and found a 2.1 % overall uncertainty of the Bass and Paur cross sections in the Huggins band up to about 330 nm. From this, $u(\delta_{o,\lambda,XS})$=2.1 % (1σ, normal distribution) is here assumed for all UVPFR wavelengths.

For the ozone amount 1 % (1σ, normal distribution) is taken as the standard uncertainty $u(\delta_{o,\lambda,DU})$. For instruments in the Canadian reference Brewer triad Fioletov et al. (2005) estimated the standard uncertainty of daily values to about 0.6 %. It was also estimated that random errors of individual observations were within ±1 % in about 90 % of all measurements. Uncertainty in ozone cross sections also introduces uncertainty in Brewer TCO determinations. Redondas et al. (2014) investigated several ozone cross section data sets and in the worst case the derived TCO differed more than 3 % from the current operational results. However, for the most recent cross section dataset (Serdyuchenko et al, 2014) that was tested, the deviation from operational values was only -0.5 % on average. Based on the expertise of the Brewer community the standard uncertainty of 1% (1σ) in total column ozone adopted here is thought to be a realistic estimate for field instruments.

The estimated uncertainty in effective ozone temperature is a function of latitude and time of the year. At low latitudes the day-to-day variation in effective ozone temperature is low. From 30° latitude and below, the uncertainty in effective ozone temperature, $u(T_o)$, is estimated to 5° C (95 % confidence level, normal distribution). At high latitudes the uncertainty is up to 10°C most of the year, with slightly lower values in June-August. For latitudes between 30°-80° the uncertainty changes from the lower to the higher values. The effect of uncertain effective ozone temperature on the ozone optical depth, $u(\delta_{o,\lambda,T})$,

is calculated as the difference between $\delta_{o,\lambda}$ at -45°C and at the temperature -45°C+$u(T_o)$, using the ozone cross section temperature dependence as described by temperature coefficients between 300-370 nm from a quadratic fit (Serdyuchenko et al., 2014).

The total standard uncertainty connected ozone optical depth is calculated as

$$u(\delta_{o,\lambda}) = \left(u(\delta_{o,\lambda,XS})^2 + u(\delta_{o,\lambda,DU})^2 + u(\delta_{o,\lambda,T})^2\right)^{1/2} \tag{12}$$

The uncertainty in the relatively small contributions from the FWHM correction of the ozone optical depth ($u(f_{o,DU})$ and $u(c_{o,\lambda,350DU})$ ) are considered to be covered in the total uncertainty based on the other ozone uncertainty terms.

**3.4.2 Uncertainty of calibration**

In the $V_0$ uncertainty, one contribution comes from the spread in the Langley plot results. The 27 Langley plot cases available from 2015 are not enough to really determine the actual distribution of the $V_0$s. It is not even possible with the additional 11 cases from 2016, to determine the actual distribution of $V_0$ for the two wavelengths, 305.3 nm and 311.3 nm, with nearly no change in sensitivity over more than a year. From the derived histograms either a normal or a triangular

distribution is plausible. The frequency distribution of $V_0$ derived with the refined Langley plot method (Eq. 2) for the 305.3 nm channel, which had the most recognisable shape, is shown in Fig. 3. As a matter of precaution, a triangular distribution is assumed for the $V_0$s at all channels, since this result in a higher standard uncertainty than if a normal distribution is used. It is hoped that this will also cover the uncertainty of the calibration method that may have been introduced by e.g. the subjective Langley event selection by the analyst. Values close to the maximum and minimum of the individual Langley plot $V_0$s,

estimated by visual inspection and where the dashed line in Fig. 3 crosses the x-axis, are taken as limits, resulting in an estimated standard uncertainty due to spread in the Langley plot $V_0$s of $u(V_{0,\lambda,L})$=[2.3 1.3 1.7 1.1]/$\sqrt{6}$ %. (Terms within brackets are here and in the following listed from the shortest to the longest UVPFR wavelength.) (See GUM (2008) for descriptions on how to calculate standard uncertainties for variables of various distributions.)

Contributing to the $V_0$ related uncertainty, there is also uncertainty added due to a possible ozone change during the Langley

plot periods not accounted for. In this respect, the systematic effect of a 0.5 DU change during each Langley plot event, made over an air mass range of 1.5, was estimated using the results in Fig. 1. This corresponds approximately to a 0.25 % mean error in the extra-terrestrial constant of the Brewer triad instruments, which is considered as a maximum value based on RBCC-E results (WMO/GAW, 2015). The values [0.7 0.3 0.1 0.05]/1.5 %, i.e. $V_0$ gradients for a 0.5 DU TCO change over an air mass range of 1.5, were taken as semi ranges of rectangular distributions for the UVPFR wavelengths, resulting

in standard uncertainties of $u(V_{0,\lambda,o})$=[0.47 0.20 0.07 0.03]/$\sqrt{3}$ %.

Also uncertainties in the $c_{FWHM}$ factors have been accounted for. The values $u(V_{0,\lambda,FWHM})$=[0.0040 0.0015 0.0005 0.0002]/$\sqrt{3}$ % were estimated in this case.

As mentioned above, in the calculations of $m_o$ for Langley plots at IZO, effective ozone altitude of 25 km was used. For ozone determination by Dobson spectrophotometers, an ozone layer altitude of about 23 km is recommended for the latitude of the IZO station (WMO/GAW, 2009). Assuming a systematically over- or underestimation of ozone altitude of 2 km (rectangular distribution), resulted in the standard uncertainty of $u(V_{0,\lambda,o3alt})$=[0.6 0.3 0.1 0.02]/$\sqrt{3}$ %.

As mentioned earlier, there is a large uncertainty in ozone optical depth at wavelengths with high ozone absorption. While this adds some uncertainty to $V_0$s of the refined Langley plot method, fortunately, the additional uncertainty in $V_0$ from the Langley method of Eq. (3) is negligible. Ozone optical depth is only used for the air mass weighting in this case.

Finally, a drift term of $u(V_{0,drift})$=1 % per year (95 % confidence level, normal distribution) has been accounted for in the total $V_0$ uncertainties. In the end, $u(V_{0,\lambda})$ is calculated as

$$u(V_{0,\lambda}) = \left(u(V_{0,\lambda,L})^2 + u(V_{0,\lambda,o})^2 + u(V_{0,\lambda,FWHM})^2 + u(V_{0,\lambda,o3alt})^2 + u(V_{0,\lambda,drift})^2\right)^{1/2}$$
(13)

Several uncertainty sources that could affect the Langley plot $V_0$s have not been taken into account due to their negligible influence. Any additional uncertainty in $V_0$ due to uncertainty in the solar position or a possible systematic effect in calculated $m_r$ is assumed to be negligible and has not been taken into account. Also, the effect of unknown vertical aerosol

distribution on the derived Langley $V_0$ was tested by assuming $m_a=m_r$, instead of the used algorithm for $m_a$. The result was only a negligible influence on the $V_0$s. The uncertainty in Rayleigh optical depth as estimated below, was calculated to affect Langley plot $V_0$s by <0.05 % and was therefore not taken into account. Finally, as discussed below (Sect. 3.4.4), any influence of circumsolar irradiance entering the FOV of the instrument has been neglected.

### 3.4.3 Uncertainty of Rayleigh optical depth

The standard uncertainty of the Rayleigh optical depth, $u(\delta_{r,\lambda})$, was derived from a 1 hPa pressure uncertainty (1σ), denoted as $u(\delta_{r,\lambda,p})$. In addition, $u(\delta_{r,\lambda,mod})$ estimated from the difference between $\delta_{r,\lambda}$ (Bodhaine et al., 1999) and the extreme values calculated for other model atmospheres by Tomasi et al. (2005, Table 5) have been taken into account. These latter differences, in the order of 0.005, were taken as limits of a 95 % confidence interval of a normal distribution. From this, the standard uncertainty of Rayleigh optical depth is estimated as

$$u(\delta_{r,\lambda}) = \left(u(\delta_{r,\lambda,p})^2 + u(\delta_{r,\lambda,mod})^2\right)^{1/2}$$
(14)

### 3.4.4 Uncertainty of measured signal including circumsolar contribution

Uncertainty in voltage readings, $u(V_\lambda)$, is calculated according to the specification of the CR10X logger for the temperature range -25°C to 50°C. The uncertainty due to additional circumsolar radiation seen within the field of view of the UVPFR, $u(V_{CS,\lambda})$, is based on the results found by Russell et al. (2004) and their Eq. 17, with coefficients A and B interpolated and

extrapolated to UVPFR field of view and wavelengths. These results are further increased by a factor of 1.25 to fit

circumsolar radiation levels modelled with the SMARTS2 model (Gueymard, 1995). These results are directly expressed as an AOD uncertainty due to circumsolar radiation in the FOV, $u(\delta_{a,\lambda,CS})$, which depends on wavelength, Ångström's wavelength exponent α and AOD amount. Therefore, the fourth term on the right hand side of Eq. (10) is replaced by $u(\delta_{a,\lambda,CS})$.

This way, the estimated additional diffuse light entering the instrument, does not result in a bias of calibration through Langley plots, since it is not dependent on air mass. In reality, as suggested by Arola and Koskela (2004), diffuse light could introduce a significant negative bias in Langley plot results at UVB wavelengths under high AOD conditions. In our case, the average UVB AOD during the Langley calibrations of the UVPFR at Izaña was only about 0.05. At the same time, the average of Ångström's wavelength exponent calculated from AOD in the 368-862 nm range was about 1.5 during the

Langley plot events, which indicates that the aerosol forward scattering was not particularly high. In addition, the maximum air mass during Langley plots never exceeded 3. It is therefore assumed that the diffuse light influence was very small on the UVPFR calibrations. Hence, this source of uncertainty was not specifically taken into account in the already conservative $V_0$ uncertainty estimation above.

### 3.4.5 Uncertainty due to neglected gaseous absorption

Absorption in $NO_2$ peaks around 400 nm but there is also some absorption at the UVPFR wavelengths, especially at the longest one. In many model reference atmospheres, the total column $NO_2$ is about 0.2 DU ($=2\cdot10^{-4}$ atm cm $=5.37\cdot10^{15}$ molecules cm$^{-2}$) (Gueymard 1995), which results in optical depths of only about $\delta_n=[0.0008\ 0.0011\ 0.0013\ 0.0019]$ at the UVPFR wavelengths. If 0.2 DU is taken as standard uncertainty of $NO_2$ amount, the approximate 95 % confidence level $NO_2$ amount becomes more than $10\cdot10^{15}$ molecules cm$^{-2}$. From OMI (Ozone Monitoring Instrument) overpass data on total

column $NO_2$ (http://www.temis.nl/airpollution/no2col/overpass_no2.html), it is concluded that at both Izaña and Davos the total column $NO_2$ should be less than $10\cdot10^{15}$ molecules cm$^{-2}$ for more than 95 % of the time. Also according to ground based measurements at Izaña the total column amount of $NO_2$ is practically always below $5\cdot10^{15}$ molecules cm$^{-2}$ (Gil et al., 2008). The $NO_2$ amount in the AERONET monthly climatology, based on the SCIAMACHY (Scanning imaging absorption spectrometer for atmospheric chartography) dataset set (http://aeronet.gsfc.nasa.gov/version2_table.pdf and references therein), is also

about 0.2 DU in Davos for the measurement periods analysed in this work. Hence, to calculate the standard uncertainty in $NO_2$ optical depth, $u(\delta_{n,\lambda})$, the $NO_2$ absorption coefficients at the UVPFR wavelengths where taken from the SMARTS2 model and multiplied by 0.2 DU $NO_2$, leading to the optical depth values mentioned above. The assumption is that uncertainty in both $NO_2$ amount and absorption coefficients, as well as in $NO_2$ airmass is included in this estimate.

At polluted sites with $NO_2$ amount frequently over 1 DU ($\approx27\cdot10^{15}$ molecules cm$^{-2}$) the influence on measured AOD

becomes significant (about 0.01 at 332 nm) and should therefore be taken into account.

For the calculation of the standard uncertainty due to neglecting absorption in $SO_2$, cross sections for $SO_2$ (valid at 195 K) determined by Vandaele et al. (1994) were used. (This data set is available at the IUP University of Bremen website http://www.iup.uni-bremen.de/gruppen/molspec/databases/dlrdatabase/sulfur/index.html.)  Brewer spectrophotometers are

also capable of measuring columnar $SO_2$ amounts. However, due to the relatively high noise levels of 1-2 DU for these measurements, they can not be used to accurately determine the normal low background $SO_2$ levels. Increased $SO_2$ levels due to e.g. volcanic eruptions are however detectable (e.g. Zerefos et al. 2017). During the UVPFR measurements at Izaña and in Davos, the co-located Brewers indeed measured average $SO_2$ values close to zero (or even slightly negative) with standard deviation <1 DU. It is therefore estimated that for the uncertainty analysis it is sufficient to use a $SO_2$ value of 0.25 DU (1σ, normal distribution) when calculating the standard uncertainty $u(\delta_{s,\lambda})$.

A $SO_2$ amount of 0.25 DU corresponds to optical depth values of about $\delta_s$= [0.0021 0.0014 0.0004 0] at the UVPFR wavelengths. At polluted sites, or when measurements are affected by a volcanic eruption ash cloud, and the $SO_2$ amount reach e.g. 2 DU, the $SO_2$ optical depth exceeds 0.016 and 0.011 at the two shortest wavelengths. Neglecting such a $SO_2$ amount introduces errors with the same order of magnitude as is connected with the $V_0$ calibration uncertainty at low airmass. It is therefore recommended to take $SO_2$ into account at least for columnar amounts of ≥2 DU.

Not taking $NO_2$ and $SO_2$ absorption and circumsolar radiation into account introduces biases in the derived $AOD_\lambda$ values. However, these biases are of different sign and therefore cancel out each other to some extent. In this example the sum of $u(\delta_{n,\lambda})$ and $u(\delta_{s,\lambda})$ equals [0.0030 0.0024 0.0017 0.0019], while $u(\delta_{a,\lambda,CS})$ is [0.0034 0.0033 0.0031 0.0029] at the UVPFR wavelengths. Still, in the calculation of the combined standard uncertainty these uncertainty sources are all added.

### 3.4.6 Uncertainty in solar position and air mass terms

Based on comparison between $R^2$ calculated in solar position algorithms by Michalsky (1988) and Reda and Andreas (2003, revised 2008) the uncertainty in sun-earth distance correction factor was estimated to $u(R^2)$=0.0003.

The actual vertical distribution of gases and aerosol particles in the atmosphere is not known. This introduces uncertainties in the relative optical air masses used for AOD calculation. As necessary input to the airmass algorithms the true or apparent solar zenith angle, $SZA_t$ and $SZA_a$, respectively, is given which is also calculated with a small uncertainty. For the UVPFR analysis the solar position algorithm by Reda and Andreas (2003, revised 2008) is used. According to the authors this algorithm is accurate within 0.0003° over eight millennia in time. This should be valid for the true solar zenith angle since the actual refraction is not known in every case. The Reda and Andreas algorithm was compared to the solar position calculations operational at PMOD/WRC for the evaluation of standard PFR measurements which utilise the solar position calculation algorithm by Montenbruck and Pfleger (1994) with refraction correction by Meeus (1991). These algorithms were always found to agree within 0.01° for tests over a number of days during different years and at different locations and altitudes. Since the UVPFR AOD determinations are limited to solar zenith angles <75°, when the differences in refraction for different atmospheric temperatures is small, the uncertainty in solar zenith angle input to air mass calculations is estimated to 0.01° (95 % confidence level, rectangular distribution).

The air mass term thought to be the least uncertain is the air mass for Rayleigh scattering, $m_r$. According to Kasten and Young (1989) their relative optical air mass formula deviates <0.07 % from more rigorous calculations at $m_r$ <7. Twice this value is taken as a 95 % confidence limit for a rectangular distribution to also take into account deviations caused by other

atmospheric conditions, mainly other vertical temperature distribution, differing from the model atmosphere used by Kasten and Young (1989). Tomasi et al. (1998) found that $m_r$ for a tropical or a 75° N summer model atmosphere differed about ±0.07 % from the Kasten and Young (1989) algorithm for $SZA_a$ up to 75°. For the total standard uncertainty $u(m_r)$ the contributions due to uncertainty in $SZA_a$ and due to algorithm uncertainty are simply added like

$$u(m_r) = 0.0014m_r/\sqrt{3} + \left(m_R(SZA_a + 0.01°) - m_R(SZA_a)\right)/\sqrt{3} \qquad (15)$$

The uncertainty in relative optical airmass for ozone is calculated by assuming that the effective ozone altitude differs ≤4 km from the used value 22 km in 95 % of the cases. So,

$$u(m_o) = \left(m_o(18\ km) - m_o(22\ km)\right)/2 \qquad (16)$$

A 4 km uncertainty (2σ, normal distribution) in the effective ozone altitude is thought to be a conservative estimate for the two sites where the UVPFR has been operating, therefore an extra contribution from a small error in true solar azimuth angle

input to the ozone airmass calculation is omitted.

In this study the vertical aerosol particle distribution is assumed to be more concentrated near the ground than the vertical distribution of the molecules of the air, leading to $m_a > m_r$. This is probably a good assumption in many situations without volcanic aerosols in the stratosphere. Nevertheless, there will be uncertainty in $m_a$ due to the unknown vertical aerosol

distribution. It is estimated that the difference between $m_a$ and $m_r$ can be taken as a 95 % confidence limit of a rectangular distribution of the uncertainty of $m_a$ due to unknown vertical distribution of the aerosol. Like for $u(m_r)$ also a contribution from $SZA_a$ uncertainty is added leading to

$$u(m_a) = (m_a - m_r)/\sqrt{3} + \left(m_a(SZA_a + 0.01°) - m_a(SZA_a)\right)/\sqrt{3} \qquad (17)$$

Not taking $NO_2$ and $SO_2$ vertical distribution into account introduces uncertainty in $m_n$ and $m_s$. For the low relative optical air masses of $<\sim 4$ considered here, it is estimated that $u(m_n)/m_a$ and $u(m_s)/m_a$ both are $<0.05$. Since $NO_2$ and $SO_2$ optical depths are assumed to be very low, as discussed above, the terms $\delta_{n,\lambda} \cdot u(m_n)/m_a$ and $\delta_{s,\lambda} \cdot u(m_s)/m_a$ are neglected in the calculation of the combined standard uncertainty of AOD.

**3.4.7 Total UVPFR AOD uncertainty**

In Fig. 4 the estimated expanded uncertainty ($U_{95}$) and the individual uncertainty components, the terms on the right hand side of Eq. (10), are shown for an example case over the airmass range 1–3.8. Both the expanded uncertainties as well as the

individual uncertainty values are given for an approximate level of confidence of 95 % in the figure. Calculations are made for measurements near sea level and a total column ozone amount of 350 DU. As a matter of precaution the AOD uncertainties are shown for a more turbid case, than the low AOD average conditions during the measurements in Davos presented below. The AOD values used at the four wavelengths are given in the graphs of Fig. 4. These corresponds to the

parameters $\alpha$=1.3 and $\beta$=$AOD_{1000nm}$=0.040 in the Ångström power law. The resulting UV AOD values are about twice as high as the mean AOD values during the measurements in Davos. Also, $\alpha$ is a bit lower than the average of about 1.5 (determined over the 368-862 nm wavelength range) during the analysed measurements not to underestimate the uncertainty due to circumsolar irradiance seen within the field of view of the UVPFR.

Clearly, the dominant part of the AOD uncertainty is caused by the uncertainty in the ozone optical depth at the three

shortest wavelengths. As the absorption by ozone decreases with wavelength the size of the $u(\delta_{o,\lambda})$ uncertainty also strongly decreases. For the longest wavelength the major contribution at low airmasses comes from the calibration uncertainty in this analysis. This is also the source of uncertainty with the strongest airmass dependence due to the $1/m_a$ reduction factor.

Major contributions to these uncertainties come from (unknown) systematic effects. Therefore, the uncertainty of average AOD values based on a number of measurements, $N$, does not decrease as much as with the factor $1/\sqrt{N}$.

It is believed that the most dominant uncertainties have been included in the current analysis. However, in addition to neglecting the effect of correlated variables, there are still some uncertainty sources which have not been taken into account when calculating the total uncertainty . For example, no information on potential non-linearity in the voltage output from the UVPFR has been found. This source of uncertainty is assumed to be small and has therefore been neglected. The pointing accuracy is monitored with the UVPFR. Normally, the pointing error is ≤0.2°. Any uncertainty caused by 0.2° pointing error

has not been taken into account. Probably more importantly, no uncertainty contributions from potential errors in the used spectral response functions have been taken into account.

For the two shortest wavelengths the estimated AOD uncertainties are very high, which of course is not very encouraging. At the same time, the estimated 2σ uncertainty at 305 nm is still only about half of estimates by Kazadzis, et al. (2005) who estimated 1σ uncertainty at UVB wavelengths to 0.07. It is therefore considered useful to continue working on AOD even at

305-306 nm to learn more on AOD retrievals in the UVB. Probably, better input information/data will be available in the future which will reduce the AOD uncertainty. If algorithms and coefficients in the AOD calculations are standardized in a network of stations, which will be the case within e.g. EUBREWNET (http://rbcce.aemet.es/eubrewnet), the precision of derived AOD values will still be high for well-maintained measurements.

## 4. Results

After the calibration at Izaña in summer 2015 the UVPFR has been operated about two months during autumn 2015 and spring 2016, respectively, at PMOD/WRC in Davos. These measurements were analysed to show an example of AOD determination with the UVPFR. The calibration results from 2015 have been used for the whole period in Davos.

As a first result cloud screened 1-minute AOD values from UVPFR #1001 during the day 12 October 2015 in Davos are shown in Fig. 5. AODs from PFR-N24 (wavelengths 368, 412, 500 and 862 nm), are also shown in the figure. During this day the turbidity in Davos was very low, which rather frequently occurs at high altitude stations. Under these conditions the effect of the FWHM corrections of the UVPFR data becomes extra important. From around 9:30 UTC, the near infrared to

the UVB range AOD increases with decreasing wavelength, according to the results in Fig. 5. Without the FWHM corrections this would not have been the case in the UV. For the whole day, AOD at 305 nm would have been lower than at 332 nm and often even lower than at 368 nm. AOD at 311 nm would also have been lower than at 332 nm part of the day. Based on these results for low turbidity conditions it is assumed that AOD from the UVPFR really do become more realistic when the proposed FWHM corrections are applied.

Daily means of cloud screened 1-minute AOD values at the 305 nm and 332 nm wavelengths are shown to the left in Fig. 6. The averages of the logarithm of daily mean AODs at all the UVPFR wavelengths, as well as from the PFR-N24, are shown to the right. Clearly, very low values of AOD are often experienced over Davos, even at UVB wavelengths. Especially during autumn 2015 this was the case. At the end of October and in November AOD at 305 nm were mostly measured lower than at 332 nm by up to 0.02 units of optical depth. In spring 2016, the turbidity conditions were higher and more variable.

The average AOD values for the whole period were measured lower than 0.1 at all four UVPFR wavelengths. While the average of daily AOD was highest at the shortest UVPFR wavelength, the average of the logarithm of daily values was actually smallest at the 305 nm wavelength due to the many very low values in autumn of 2015 which get more weight when using the logarithm of the AOD.

During the measurements in Davos the average AOD values in the UVB are not very well estimated by extrapolating AOD

values at the UVA-NIR wavelengths using the common Ångström relation, represented by the (red) full line in the right panel of Fig. 6. To be more specific, extrapolated AOD at UV wavelengths is overestimated. Using a second order fit in the log-log space, earlier introduced by Eck et al. (1999), leads to better results, at least for the two longest UVPFR wavelengths. As shown above, the uncertainties of the UV AOD values are however considerable and the AOD values measured by the UVPFR are not significantly different from any of the extrapolated values in this low turbidity case.

In the calibration section above (Sect. 2.2) the UVPFR sensitivity was shown to be satisfactorily stable over one year. As an additional stability and consistency check, AOD from the UVPFR has been compared to AOD derived from a Brewer spectrophotometer. At PMOD/WRC, the Brewer Mk III #163 is operated. This instrument provided the ozone values used in the AOD calculations based on spectral transmission data from the UVPFR in Davos.

Using the calibration against the UVPFR during the RBCC-E campaign in 2015, UV AOD has been determined from

Brewer #163 during its measurements in Davos. Also a small temperature correction was applied to the Brewer direct irradiance readings as well as a polarization correction suggested by Cede et al. (2006).

The comparison of AOD from Brewer #163 and the UVPFR #1001 in Davos is shown in Fig. 7. Since the UVPFR has the highest sampling rate (1 measurement/minute) UVPFR AODs were first interpolated (linearly) to Brewer DS measurement

times. These UVPFR AOD values at Brewer DS times were then further interpolated from the nearest surrounding UVPFR wavelength pair to the Brewer wavelengths using the Ångström relation.

Individual AOD differences (UVPFR-Brewer) for cloud screened and near simultaneous measurements are shown in Fig. 7. In the graphs, also the suggested WMO traceability limits for absolute AOD differences (that have been defined for AOD at wavelengths without gaseous absorption in the UVA-NIR wavelengths range) (WMO/GAW, 2005) are shown. Obviously, the agreement is very good between the Brewer and the UVPFR for these measurements taken 4-11 months after the calibration. During the calibration of Brewer #163, more than 98 % of the AOD residuals, AOD(Brewer)-AOD(UVPFR), were within the WMO limits at all wavelengths. During the comparison in Davos, at four out of the five Brewer wavelengths, more than 95 % of the differences fall within the WMO limits. Only at the shortest wavelength, with 85.6 % of the differences within the limits, the traceability requirement of 95 % was not fulfilled. This could indicate a small change in any of the instruments at the shortest wavelength(s). The root mean squared difference is still low at all wavelengths, amounting to [0.008, 0.006 0.006 0.005 0.005] for the 306-320 nm wavelengths.

During the low AOD period from end of October until November, also AOD from the Brewer showed the unexpected behaviour of giving decreasing AOD values with decreasing wavelength. Therefore, the AOD differences between the UVPFR and the Brewer remained small also during this period. There are several possible explanations for the low AOD values at the shortest wavelengths. The most plausible reason should be that the used ozone absorption coefficients and/or ozone amount were too high. Also, the use of too low calibration values could be a possible contributor. Based on the relatively stable differences over the day between AOD at e.g. 305 nm and 368 nm, in addition to the fact that $V_0$ for the 305 nm channel would need to be increased by ≥2 % to give expected AOD values, it is believed that erroneous calibration is not the major issue.

## 5. Conclusions

This paper reports on the UVPFR sunphotometer, an instrument that can be used for AOD measurements at four UV wavelengths. The standard PFRs were designed with emphasis on precision and stability, while also being robust instruments. These goals have been reached by the PFRs (Wehrli 2000; Gröbner et al., 2015). The UVPFR is of similar design and based on the results of this first study, including suggested corrections, the UVPFR appears to be a stable high quality radiometer for AOD determination in the UV. According to Langley plot calibrations at a high altitude station the sensitivity of the UVPFR changed by ≤1.1 % over a 13-14 month period.

It was shown that due to the relative wide FWHM of the UVPFR the calibration constants ($V_0$) from Langley plot calibrations underestimate the true extra-terrestrial signals. Accordingly, correction factors were suggested. The effect of the finite FWHM is an apparent wavelength shift towards longer wavelengths as airmass increases, especially for the shorter UVPFR wavelength channels 305 nm and 311 nm. This also results in an apparent decrease in ozone optical depth with

increasing airmass. An adjusted formula for the calculation of AOD with a correction term dependent on total column ozone amount and ozone air mass (Eq. 8) was therefore introduced.

Even with the suggested corrections applied, the expanded uncertainty of AOD derived from UVPFR measurements, as well as from other UVB instruments, remains relatively high at the shortest wavelengths. The major source of uncertainty is the

ozone optical depth uncertainty, resulting from uncertainties in ozone cross section, ozone temperature and TCO amount. The second largest source of uncertainty at the three shortest wavelengths, and the largest source of uncertainty at 332 nm, is the calibration uncertainty, especially at high sun/low airmass conditions.

Despite the relatively high AOD uncertainties at the short wavelengths, it is still considered worthwhile to continue working with the AOD at e.g. 305-306 nm to learn more on AOD retrievals in the UVB. Most probably, better input information

connected to ozone will be available in the future which will reduce the AOD uncertainty. Also, if the same ozone cross section data and effective ozone temperature data are used by different instruments/groups/sites, as will be the case within EUBREWNET for example, the AOD results will be consistent and much more comparable.

An example of very good agreement of UV AOD retrievals was shown by a comparison between the UVPFR #1001 and Brewer #163 for several months of measurements in Davos. Since Brewer #163 and UVPFR #1001calibrations were partly

linked at an earlier date, the comparison was not performed by fully independent instruments and therefore we should expect a relatively good agreement. The comparison indeed confirms good agreement for the measurements taken 4-11 months after the Brewer calibration. The root mean squared AOD differences were <0.01 at all the 306-320 nm Brewer wavelengths. This can be considered a very good result for an AOD comparison at UVB wavelengths. An additional very likely reason for the good agreement is the fact that both instrument types measure at close wavelengths in the UVB. In earlier studies in which

AOD was determined from Brewer direct sun measurements the validation has so far only been done against measurements at UVA or even visible wavelengths (Marenco et al., 2002; Cheymol and De Backer, 2003; Cheymol et al., 2006; Gröbner and Meleti, 2004; Kazadzis et al. 2005; Kazadzis et al. 2007; De Bock et al., 2010; Kumharn et al., 2012;). Also earlier comparisons of AOD from Brewers of different type have shown larger differences than between the UVPFR and the MkIII Brewer in this study (Kazadzis et al. 2005; Kumharn et al., 2012;).

In addition to a low turbidity case showing AOD values from the UVPFR consistent with a standard PFR, average UV AOD values of the UVPFR during the measurements in Davos were compared with highly accurate AOD values, 2σ uncertainties estimated to <0.01, at UVA-NIR wavelengths from a standard PFR. Extrapolated AODs at UVPFR wavelengths using a second order polynomial fit of $ln$(AOD) versus $ln$(λ) were closer to the mean values measured by the UVPFR than when a first order fit, i.e. the common Ångström relation, was used for extrapolation. However, in both cases the differences

between the extrapolated and the measured values were smaller than the estimated UVPFR AOD uncertainties for the low AOD conditions experienced during the measurements in Davos.

Despite the fact that the total uncertainty of AOD in the UVB is relatively high, based on the comparison between the UVPFR and a Brewer it is estimated that calibrated and well maintained UVPFR sunphotometers and Brewer spectrophotometers can measure AOD at a precision of 0.01 (1σ) at their direct sun measurement wavelengths.

**Data availability**

The total column ozone data used in this study can be downloaded from the EUBREWNET website: http://rbcce.aemet.es/eubrewnet.

**Acknowledgements**

5  T. Carlund was supported through Grant Nr. C14.0025 from the Swiss Staatssektretariat für Bildung, Forschung und Innovation (SBFI) within COST ES1207. Part of the work was supported by a STSM Grant from COST Action ES1207 (EUBREWNET – A European Brewer Network). The slit function measurements were done on the tuneable laser facility ATLAS, funded through contract number IDEAS+/SER/SUB/11. The total column ozone values from the Brewer triad at the Izaña observatory were kindly provided by Alberto Redondas at IARC/AEMET.

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

**Table 1: Wavelength characteristics of UV-PFR#1001 based on laboratory measurements February 2016. The third column shows effective central wavelength resulting from convolving the spectral response function with an extra-terrestrial solar spectrum.**

| Channel (nm) | Effective central wavelength (nm) | Convolved effective central wavelength (nm) | Bandwidth FWHM (nm) |
|---|---|---|---|
| 305 | 305.35 | 305.31 | 0.99 |
| 311 | 311.36 | 311.34 | 1.04 |
| 318 | 317.55 | 317.50 | 1.20 |
| 332 | 332.33 | 332.32 | 1.26 |

**Table 2: Langley calibration results for UV-PFR#1001 at Izaña 2015 and 2016, together with calculated $V_0$ and $\delta_o$ FWHM correction factors. Also the used Rayleigh optical depth and ozone absorption coefficients used for the UV-PFR#1001 are given.**

| Channel (nm) | Mean $V_0$ 2015 (mV) | Std.dev. of $V_0$ (Std.dev of mean $V_0$) 2015 (%) | Mean $V_0$ 2016 (mV) | $V_0$ change 2015–2016 (%) | FWHM correction factor for V0 $c_{FWHM}$ | $\delta_o$ corr. factor at 350 DU, $c_{o,350DU}$ | $\delta_{R,\lambda}$ Bodhaine (1999) | $k_{o,\lambda}$ B&P (-45°C) cm$^{-1}$ |
|---|---|---|---|---|---|---|---|---|
| 305 | 30319 | 1.28 (0.25) | 30257 | -0.2 | 1.012 | -0.0045 | 1.1287 | 4.4682 |
| 311 | 11531 | 0.70 (0.13) | 11522 | -0.1 | 1.003 | -0.0010 | 1.0377 | 2.0362 |
| 318 | 10669 | 0.82 (0.16) | 10553 | -1.1 | 1.001 | -0.0004 | 0.9542 | 0.8802 |
| 332 | 5302 | 0.44 (0.08) | 5248 | -1.0 | 1.000 | 0 | 0.7856 | 0.0597 |

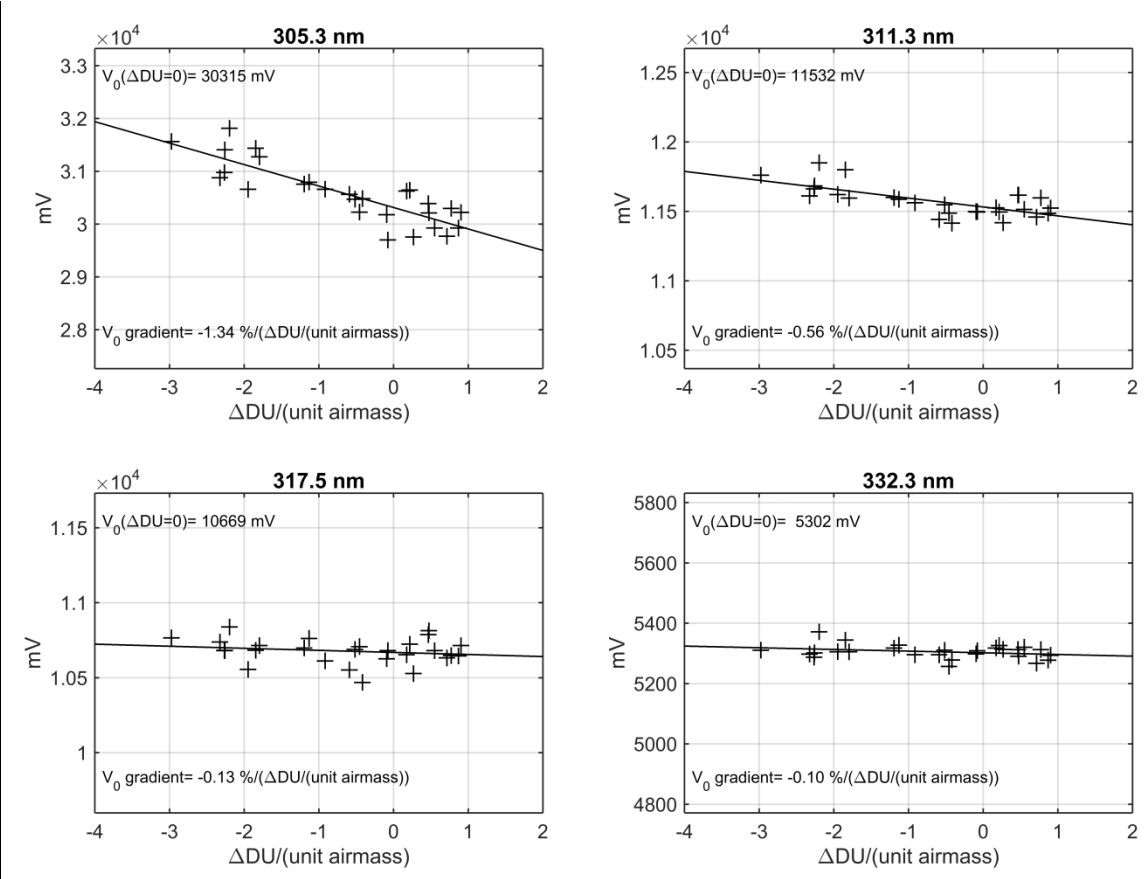

**Figure 1: Results of all the Langley plot calibrations at IZO during May-August 2015. The final $V_0$s are derived from linear interpolation at zero ozone change. The ozone change during each Langley event is calculated from a linear fit of the Brewer triad total ozone values versus ozone airmass during the Langley event.**

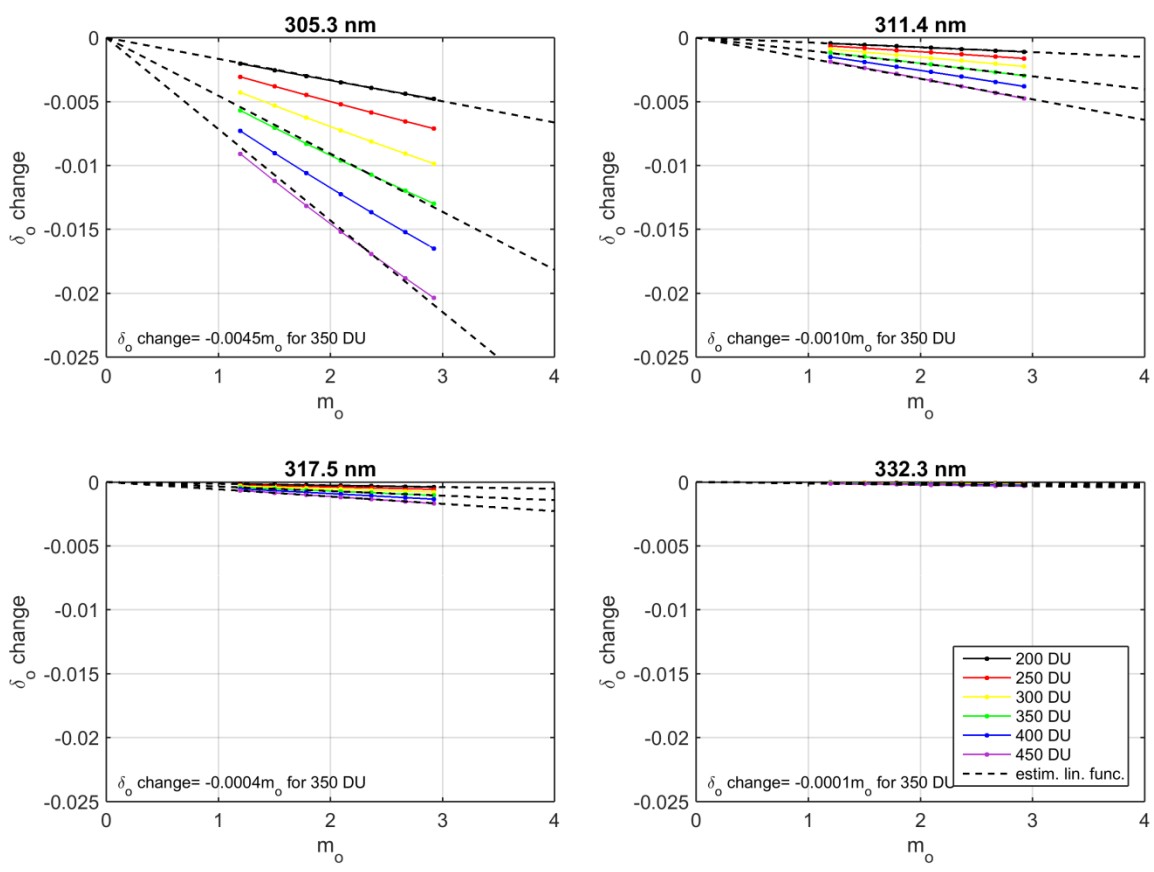

**Figure 2: Calculated change in effective ozone optical depth with airmass due to the UVPFR filter bandwidths.**

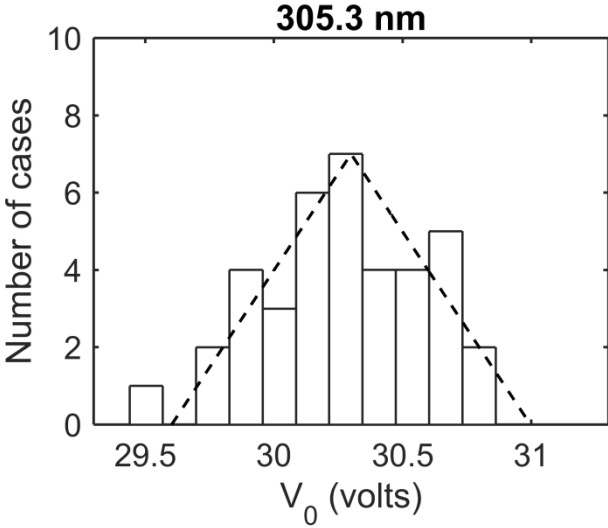

**Figure 3. Frequency distribution of V0 for the 305 nm channel derived with the refined Langley plot method (Eq. 3) during both calibration periods 2015 and 2016 at Izaña. The results were approximated with a triangular distribution indicated by the dashed line.**

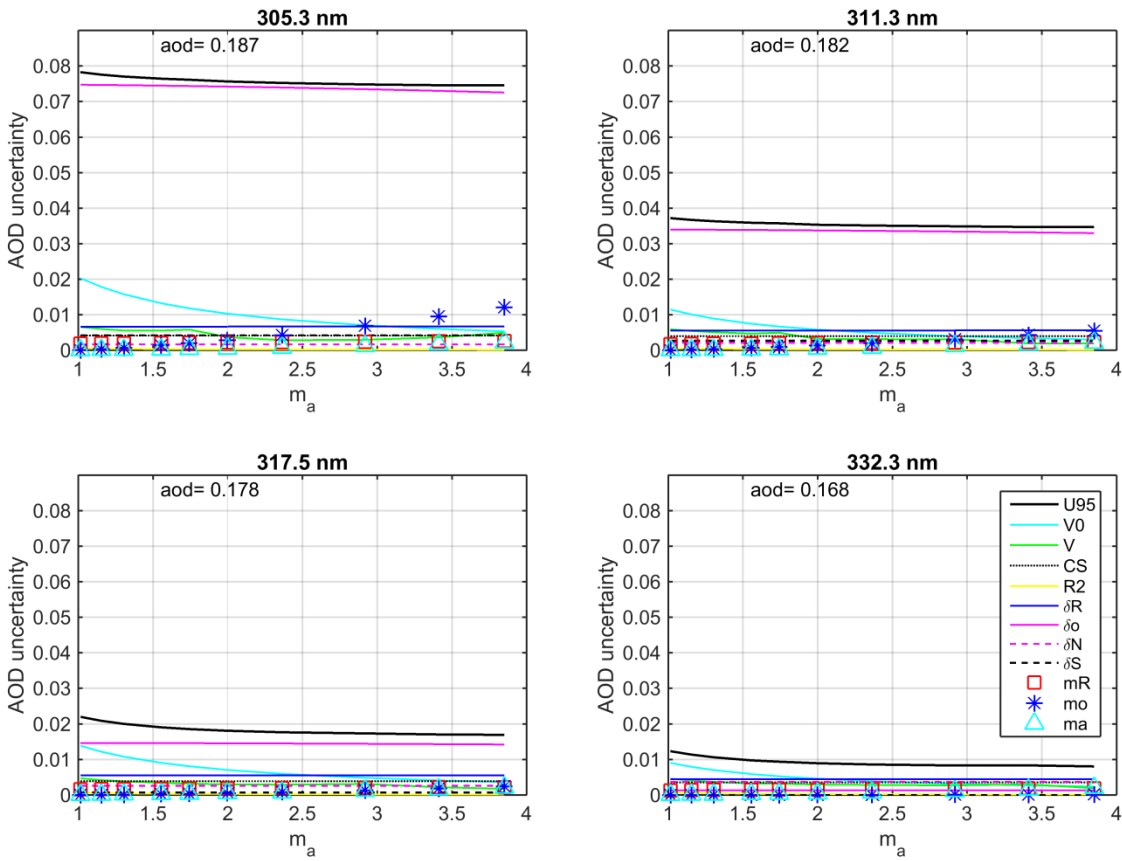

**Figure 4: Estimated expanded uncertainty, k=2, (black lines) of AOD for the UVPFR #1001 wavelengths. Individual contributing uncertainties sources, at an approximate level of confidence of 95 %, are also shown. Calculations are made for a day 2 months after a calibration and with total column ozone amount of 350 DU.**

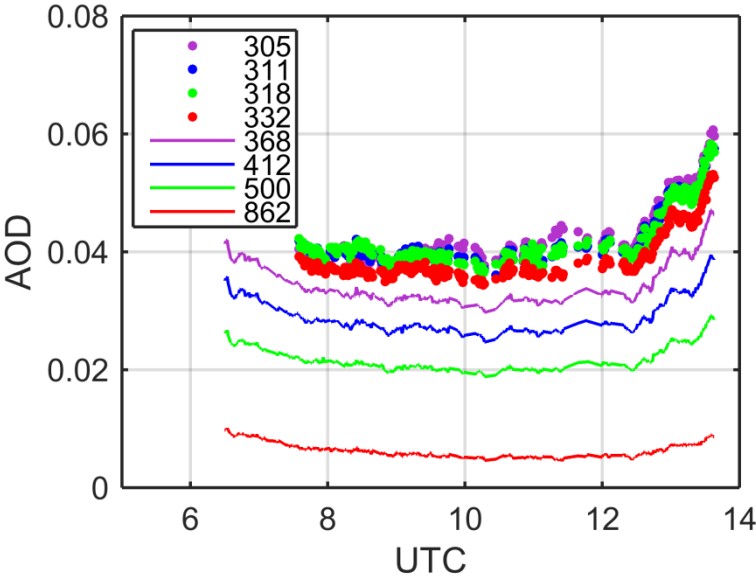

**Figure 5: 1-minute AOD determined by UVPFR #1001 (dots) and PFR-N24 (lines) on the 12th October 2015 in Davos. Data points disturbed by clouds have been removed.**

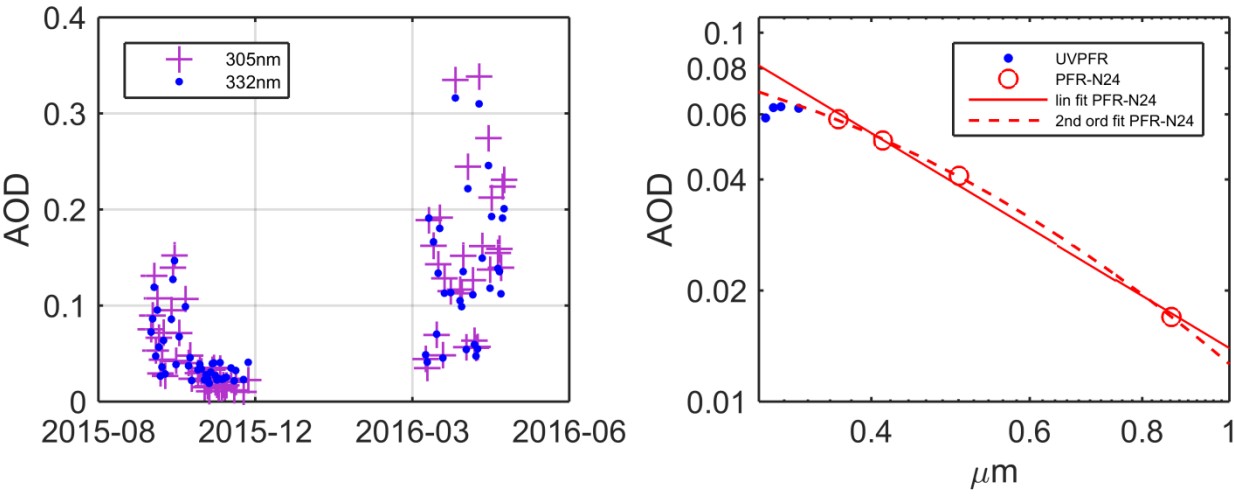

**Figure 6: Daily mean AOD at 305 and 332 nm in Davos (left) and mean of daily means of AOD during the whole study from the UVPFR and a standard PFR (right).**

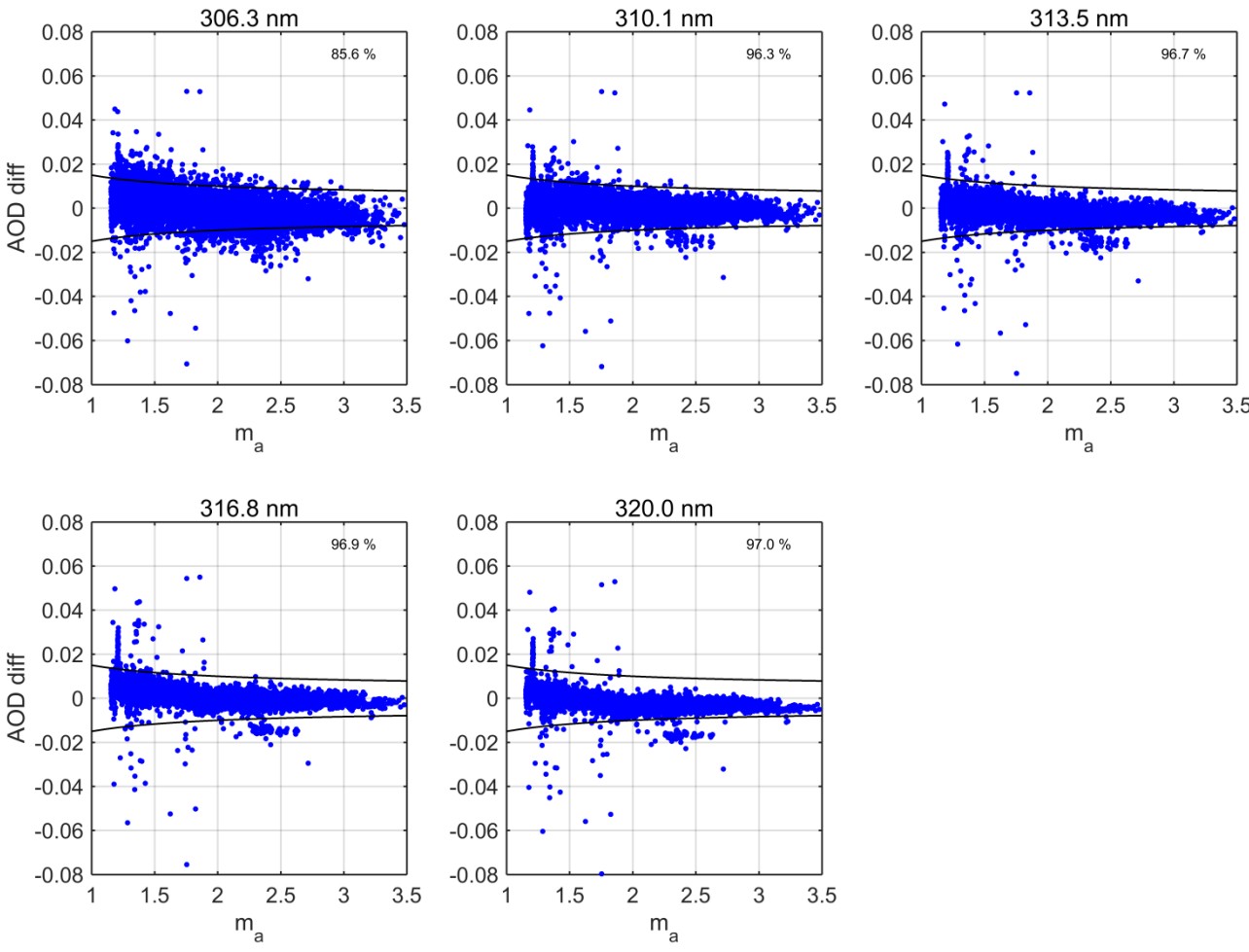

**Figure 7: Differences in AOD, UVPFR-Brewer, at Brewer wavelengths for measurements during autumn 2015 and spring 2016 in Davos. Percentage of differences within WMO traceability limits is given in each graph.**