# Peer review of "Aerosol optical depth determination in the UV using a four-channel precision filter radiometer"

_Atmospheric Measurement Techniques, 2016_

## Referee Comment (RC1) · Anonymous Referee #2 · 11 Jan 2017

General comments

Carlund et al. describe the performances of a UVPFR photometer in measuring aerosol optical depths in the UV (including UV-B) region, a very interesting topic addressed in only few other publications. The paper is well written and the authors clearly explain every step of the analysis (from the calibration to the measurements), describe the potential error sources (as from the finite FWHMs of the instrument) and provide an exceptionally comprehensive discussion about the uncertainty of aerosol optical depth retrievals. The stability of the instrument is proven by Langley calibrations performed about one year apart and through comparison with a Brewer spectrophotometer. The paper fits well within the scope of AMT and only minor corrections are necessary, in

my option.

I have some general comments to improve the readability:

1. Structure of the paper. A more conventional naming of the sections (e.g. "2. Instruments and measuring sites", "3. Methods", etc.) would improve the readability. The description of the Izaña observatory (page 4, lines 6-19) could be moved in a dedicated paragraph. Also, the Davos site should be briefly described. Furthermore, the Brewer spectrophotometer should be shortly illustrated for unexperienced readers in the "Instruments" section. An additional section should also be reserved for the assessment of the stability using the Brewer spectrophotometer (page 16 lines 8-29).

2. The formulae are not entirely commented and further explanation of the terms should be added (cf. "Specific comments" and "Technical corrections").

Specific comments

- page 5 line 16: "errors in d0". Isn't (d0+da) retrieved from the fit in Eq. 3? What kind of error are you referring to?

- page 6 line 15: "due to variation with wavelength in extra-terrestrial solar irradiance and ozone..."clashes with "the FWHM effects ... are entirely caused by the rapidly increasing ozone absorption with decreasing wavelength" (page 8 line 5). Please, explain if the first sentence may refer to larger FWHMs or if the spectral variation of the extra-terrestrial solar irradiance was only a potential, but not real, issue;

- page 6 line 28: explain why you chose these values for the Angstrom parameters. Are they representative of a climatology? (They are also repeated on page 14 line 24)

- page 7 line 24: write formula for ozone optical depth correction. Anyway, wouldn't it be more physical to consider the slant column density (SCD) of ozone instead of ozone vertical column (VCD) and airmass separately?

- page 8 line 25: explain why two different sets of ozone cross sections and effective

ozone temperatures are used throughout the paper (Bass&Paur in Sect. 3 and IUP in Sect. 2.3);

- page 10 line 14: explain why 1% was chosen for the ozone amount uncertainty. References?

- page 10 Sect. 4.2: are uncertainties from the Rayleigh optical depth (Sect. 4.3), circumsolar radiation (Sect. 4.3), neglected aerosol absorption (Sect. 4.5) and solar position (Sect. 4.6) expected to impact on V0? In that case, include them in the discussion;

- page 10 line 30: a figure of the V0 distribution could be interesting for the reader, even though it is not fitted well by any known statistical function;

- page 12 line 20: "The amount of NO2 in the atmosphere over the measuring site is unknown". Is it true for the calibration site as well? Are there no NO2 measurements at IZO?

- page 15 line 31: notice that the Angstrom relation is not linear, therefore averages should be in principle calculated on log(AOD) instead of AOD to obtain a straight line from log(AOD) vs log(lambda) as in Fig. 5b;

- page 16 line 27: "with 85.6% of the differences within the limits... this indicates a small change". Could you write what percentage was reached at El Arenosillo at this wavelength during calibration transfer?

Technical corrections

- page 1 lines 28-30: the sentence is a duplicate of the text written just after (page 2 lines 4-9);

- page 2 line 1: carify that the "expected increase of UV radiation to to the declining ozone levels" applies to past series;

- page line 11: avoid repetition "provides... provide...";

- page 1 line 24-25: "based on Brewer AOD retrievals, ... related direct sun retrieval for AOD". Redundant?

- page 1 line 28: "PMOD/WRC": define acronym here, not on page 3 line 4;

- page 3 lines 17-19: "For the filter bandwidths, ..." it is not clear how the two following sentences are connected to the text and what are the results of the determinations by the tunable laser;

- page 4 line 20: "V0" should be maybe repeated after the comma;

- page 5 line 1: "replaces" –> "replace";

- page 5 Eq. 2: why the "Rˆ2" factor appears starting only from Eq. 2 and not 1?

- page 5 line 10: "on" –> "of";

- page 5 line 11: clarify what "in these cases" means. You are probably saying that the ozone and aerosol measurement series are useful to accurately calculate the weighting sum m2ODw?

- page 6 line 19: remove "Numbers on";

- page 6 line 20: "exponential" is not rigorous here;

- page 7 line 10: write the formula for correction factors;

- page 7 line 16: "if not a further" –> "if no further";

- page 8 Eq. 5 "Rˆ2" should be on the left-hand side;

- page 9 line 1: "more recent Rayleigh scattering coefficients". Specify what coefficients (Bodhaine?);

- page 9 Eq. 7: define what f is in the equation;

- page 9 Eq. 8: "V_CS" included in the equation is explained only 3 pages later in the text (Sect. 4.4). Please, add a sentence after Eq. 8 explaining that this variable will be

commented later.

---

## Referee Comment (RC3) · Anonymous Referee #1 · 14 Jan 2017

This paper deals with the determination of the aerosol optical depth at four UV wavelengths using a UVPFR sunphotometer, a modified version of the Precision Filter Radiometer. Details concerning the instrument's characteristics, calibration and performance are presented and a deep analysis for the uncertainties in the AOD estimations takes place. The manuscript is well written with sufficient background literature. The used methodologies and analyses are appropriate, support the findings of the study and the presentation of the results is clear. Since aerosols are of great importance in various atmospheric issues, I think that the UVPRF instrument would be a valuable tool for the scientific community. I believe that the study fulfills the scope of ATM and I recommend to be published after minor revisions.

[Figure]

Comments/suggestions: A sensitivity analysis for SO2 and NO2 would be significant information about instrument's performance at polluted places.

Page 5, line 15-16: "On the other hand, based on numerical tests, V0 results using equation 3 were found less sensitive to errors in $\delta$o". Is there any explanation for this? Page 5, line 22: I suggest the results of V0s in 2016 to be added in Fig.1 using different symbols. Page 7, line 9: The average conditions should be reported. Page 8, line 1: Specify the Rayleigh scattering coefficients used for the ozone calculation. Page 8, line 10: The modifications on the cloud screening should be reported. Page 16, line 23: Reference for the traceability limits for absolute AOD differences should be given.

Technical comments Page 1, line 29: humans instead of Humans Page 2, line 17: Check the format of word Only Page 8, line 18: According to equation 6

Missing citations: Page 1, lines 29-30: Madronich et al., 2015; UNEP, 2010 Page 2, lines 1-2: Zerefos et al., 2012; De Bock et al., 2014 Page 2, line 11: Holben et al., 1998 Page 2, lines 22-23: Meleti et al., 2009; Kumharn et al., 2012 Page 4, line 32: Schmid and Wehrli, 1995 Page 7, line 4: McArthur et al., 2003 Page 12, line 4: Russel et al., 2004 Page 13, line 4: Zerefos et al., 2016

References not cited in the manuscript Page 19, lines 8-9: Cede, A., Labow, G., Kowalewski, M., and Herman, J.: The effect of polarization sensitivity of Brewer spectrometers on direct Sun measurements, in: Ultraviolet Ground- and Space-Based Measurements, Models, and Effects IV, Proc. SPIE Int. Soc. Opt. Eng., 5545, 131–137, 2004. Page 20, lines 11-13: Eck, T. F., Holben, B. N., Reid, J. S., Dubovik, O., Smirnov, A., O'Neill, N. T., Slutsker, I., and Kinne, S.: Wavelength dependence of the optical depth of biomass burning, urban and desert dust aerosols, J. Geophys. Res., 104, D24, 31333– 31349, 1999. Page 20, lines 19-20: Fioletov, V. E., Kerr, J. B., McElroy, C. T., Wardle, D. I., Savastiouk, V., and Grajnar, T. S.: The Brewer reference triad, Geophys. Res. Lett., 32, L20805, doi:10.1029/2005GL024244, 2005. Page 22, lines 21-22: McPeters, R.D. and Labow, G.J.: Climatology 2011: An MLS and sonde derived

ozone climatology for satellite retrieval algorithms, J. Geophysical Res., 117, D10303, doi:10.1029/2011JD017006, 2012. Page 23, lines 4-5: Nicolet, M.: On the molecular scattering in the terrestrial atmosphere: An empirical formula for its calculation in the homosphere. Planet. Space Sci., 32, 1467–1468, 1984. Page 23, lines 11-13: Redondas, A., Evans, R., Stuebi, R., Köhler, U., and Weber, M.: Evaluation of the use of five laboratory-determined ozone absorption cross sections in Brewer and Dobson retrieval algorithms, Atmos. Chem. Phys., 14, 1635-1648, doi:10.5194/acp14-1635-2014, 2014.

Figure 1: Use subscript character for 0 in the parameter V0 - Figure caption: Langley event instead of Langley episode Figure 2: Use the same Y-axes scale for the 4 graphs. I suggest the word the truth to be replaced on the legend. Figure 5: Use different symbols for the AOD at 332 nm (or at 305 nm).

———————————————

---

## Author Comment (AC1) · 16 Feb 2017

Manuscript for AMT:

**Aerosol optical depth determination in the UV using a four-channel precision filter radiometer**

by T. Carlund, N. Kouremeti, S. Kazadzis, and J. Gröbner

**Responses to Anonymous Referee #1**

(Reviewer comments in *italic*.)

*Comments/suggestions: Sensitivity analysis for SO2 and NO2 suggested.*
Comments about sensitivity to $NO_2$ and $SO_2$ under more polluted atmosphere have been added.

*Page 5, line 15-16: On $V_0$ results using Eq. 3 found less sensitive to errors in $V_0$:*
An explanation has been added.

*Page 5, line 22: Add $V_0$ results from 2016 in Fig. 1.*
It was suggested to add also $V_0$ results from 2016 to Figure 1. When doing so, the main purpose of the figure, showing the different sensitivity to (measured) ozone change during Langley events at different wavelengths, becomes less clear due to the change in sensitivity from 2015 to 2016, especially at the two longer wavelengths. Therefore, we think showing $V_0$ values only for 2015 is a better choice and the results from 2016 have not been added to the figure.

*Page 7, line 9: Specification of average conditions during Langley calibrations requested.*
Numbers on average conditions of air mass range and TCO during Langley calibrations have been added.

*Page 8, line 1: Specify Rayleigh scattering coefficients.*
It is assumed that this comment is for page 9, line 1. The Rayleigh scattering coefficients have been specified.

*Page 8, line 10: Modifications of the Alexandrov (2004) cloud screening should be reported.*
It is assumed that this comment is for page 9, line 10. The modifications of Alexandrov et al. (2004) cloud screening have been described.

*Page 16, line 23: Reference for the traceability limits for absolute AOD differences should be given.*
Reference has been added.

Technical comments

*Errors at page 1 (line 29), page 2 (line 17) and page 8, supposedly page 9 (line 18):*
Errors have been corrected.

*Missing citations:* The missing references have been added. (De Bock et al., 2014 and Meleti et al., 2009, were already in the first manuscript.) More references have been added during the manuscript revision process.

*References not cited in the manuscript:* References not cited in the manuscript have been removed. Eck et al. (1999), Fioletov et al. (2005), Nicolet (1984) and Redondas et al. (2014) are now referred to in the text.

*Figure 1.* Suggested corrections applied.

*Figure 2.* Suggested corrections applied.

*Figure 5.* Suggested correction applied. While working on this figure it was realized that the data plotted in the left panel of figure 5, as well as in figure 4, was of the wrong version. This has been corrected. Unfortunately, this had the effect that, especially at very low AODs, often AOD at 305 nm became smaller than AOD at 332. Part of text in the result section therefore had to be revised.

---

## Author Comment (AC2) · 16 Feb 2017

Manuscript for AMT:

**Aerosol optical depth determination in the UV using a four-channel precision filter radiometer**

by T. Carlund, N. Kouremeti, S. Kazadzis, and J. Gröbner

**Responses to Anonymous Referee #2**

(Reviewer comments in *italic*.)

General comments

1. *Structure of the paper. A more conventional naming of the sections (e.g. "2. Instruments and measuring sites", "3. Methods", etc.) would improve the readability. The description of the Izaña observatory (page 4, lines 6-19) could be moved in a dedicated paragraph. Also, the Davos site should be briefly described. Furthermore, the Brewer spectrophotometer should be shortly illustrated for unexperienced readers in the "Instruments" section. An additional section should also be reserved for the assessment of the stability using the Brewer spectrophotometer.*
   The sections in the manuscript have been rearranged and, in some cases, renamed according to the recommendations. Descriptions of the sites have been added, as well as a short description of the Brewer. Since the Brewer spectrophotometer is not in the main focus of the paper, an extensive review on Brewer stability has not been given. But the results of operational stability monitoring of the used Brewer #163 during the period analysed are given.

2. *The formulae are not entirely commented and further explanation of the terms should be added.*
    Explanations of terms in formulas have been improved.

Specific comments

*Page 5, line 16: "errors in d0". Isn't (d0+da) retrieved from the fit in Eq. 3? What kind of error are you referring to?*
Attempt has been made to clarify the text.

*Page 6, line 15: "due to variation with wavelength in extra-terrestrial solar irradiance and ozone..."clashes with "the FWHM effects … are entirely caused by the rapidly increasing ozone absorption with decreasing wavelength" (page 8 line 5). Please, explain if the first sentence may refer to larger FWHMs or if the spectral variation of the extra-terrestrial solar irradiance was only a potential, but not real, issue.*
The first sentence has been corrected.

*Page 6, line 28: Explanation the choice of Ångström parameters.*
Explanation for the choice of Ångström parameters added.

*Page 7, line 24: write formula for ozone optical depth correction. Anyway, wouldn't it be more physical to consider the slant column density (SCD) of ozone instead of ozone vertical column (VCD) and airmass separately?*

Formula for ozone optical depth correction is added a bit later in the text (new Eq. 6). It is a good point that slant column ozone density ($SCD=m_o \cdot$TCO) could be used for the ozone optical depth correction, instead of treating $m_o$ and TCO separately. However, maybe in contrast with intuition, it turned out that even a correction based on SCD would have different sensitivity for different ozone amounts, even though the difference is not as large as for the current case, shown in Fig. 2. Anyway, a similar expression, i.e. $\Delta\delta_{o,\lambda}= f'_{o,DU} \cdot c'_{o,\lambda,350DU} \cdot$SCD, would have been needed also in this case to get an accurate correction. Therefore, the current formulation has been kept.

*Page 8, line 25: Explain why two different sets of ozone cross sections and effective ozone temperatures are used throughout the paper (Bass&Paur in Sect. 3 and IUP in Sect. 2.3).*
IUP ozone cross sections were used in the modelling of the FWHM effects due to the fact that they were available at the same resolution as the used extra-terrestrial solar spectrum. This has been clarified in the text. Bass & Paur (1985) cross sections are used in the operational TCO determinations and therefore they were also used for AOD determinations with Brewers, as well as for the UVPFR. This has also been clarified in the text.

*Page 10, line 14: Explain why 1% was chosen for the ozone amount uncertainty. References?*
Explanation for the choice of 1 % TCO uncertainty, including some references, has been added.

*Page 10, Sect. 4.2: Are uncertainties from the Rayleigh optical depth (Sect. 4.3), circumsolar radiation (Sect. 4.3), neglected aerosol absorption (Sect. 4.5) and solar position (Sect. 4.6) expected to impact on V0? In that case, include them in the discussion.*
This section has largely been rewritten and more potential $V_0$ uncertainty sources are discussed. Neglected aerosol absorption, as suggested in the review comments, should not have any influence on Langley V0s and is therefore not mentioned.

*Page 10, line 30: a figure of the V0 distribution could be interesting for the reader, even though it is not fitted well by any known statistical function.*
A figure of $V_0$ distribution at one of the two wavelengths with small sensitivity change from 2015 to 2016 has been added.

*Page 12, line 20: "The amount of NO2 in the atmosphere over the measuring site is unknown". Is it true for the calibration site as well? Are there no NO2 measurements at IZO?*
Good point. $NO_2$ measurements are indeed performed at IZO. The text in Sect 4.5 has been changed.

*Page 15, line 31: notice that the Angstrom relation is not linear, therefore averages should be in principle calculated on log(AOD) instead of AOD to obtain a straight line from log(AOD) vs log(lambda) as in Fig. 5b.*
Good point. Change to average of log(AOD) has been made in text and Fig. 5, right panel. Implications of the change are commented.

*Page 16, line 27: "with 85.6% of the differences within the limits: : : this indicates a small change". Could you write what percentage was reached at El Arenosillo at this wavelength during calibration transfer?*
Percentage of AOD differences/residuals at El Arenosillo calibration added.

Technical corrections

Page 1, lines 28-30: The first paragraphs of the Introduction have been rewritten.

Page 2, line 1: Suggested clarification adopted.

Page 2, line 11: Suggested correction applied.

Page 1, lines 24-25: It is assumed that this comment is for page 2, lines 24-25. Sentence corrected.

Page1, line 28: It is assumed that this comment is for page 2, line 28. Acronym defined here.

Page 3, lines 17-19: It is assumed that this comment is for lines 22-24. Text has been rewritten.

Page 4, line 20: Suggested correction applied.

Page 5, line 1: Suggested correction applied.

Page 5, Eq. 2: $R^2$ inserted in also Eq. 1, and explanation given after Eq. 1.

Page 5, line 10: Suggested correction applied.

Page 5, line 11: Sentence has been rewritten.

Page 6, line 19: Suggested correction applied.

Page 6, line 20: The somewhat careless use of "exponential" has been removed.

*Page 7, line 10: Write the formula for correction factors.*
It is not exactly understood what formula is requested but the following formulation has been added:
"… the $V_0$ correction factors $c_{FWHM} = V_{0,\lambda,true}/V_{0,\lambda}$(Langley) were estimated to $c_{FWHM}$=[1.012 1.003 1.001 1.000] for the UVPFR channels…"

Page 7, line 16: Suggested correction applied.

*Page 8, Eq. 5: "Rˆ2" should be on the left-hand side.*
Suggested correction applied. (Thanks!)

Page 9, line 1: The used Rayleigh scattering coefficients have been specified.

*Page 9, Eq. 7: Define what f is in the equation.*
"*f*" has been defined.

Page 9, Eq. 8: Suggested correction applied.

---

## Author Comment (AC3) · 16 Feb 2017

Manuscript for AMT:

**Aerosol optical depth determination in the UV using a four-channel precision filter radiometer**

by T. Carlund, N. Kouremeti, S. Kazadzis, and J. Gröbner

**Responses to Anonymous Referee #3**

(Reviewer comments in *italic*.)

General comments

1. *Please make sure that you explain all the acronyms/abbreviations the first time they are used in the text.*
   Manuscript has been corrected so that acronyms and abbreviations are explained at their first appearance.
2. *Some of the equations could do with a bit more explanation on the used terms.*
   Explanations of terms in equations have been improved.

Scientific comments

*Page 1-2, Introduction: The terms UV and UVB are not always used consistently in the introduction. I would also specify the wavelength range of UVA and UVB somewhere in the text.*
UVA and UVB have been specified in the Introduction. Introduction has partly been rewritten.

*Page 1, line 28: You write in line 28 that the 'absorption' of surface UV by aerosols has become of major interest because of the harmful effects on UV on humans and the biosphere. I would suggest to write 'extinction' as aerosols can also scatter UV radiation hence increasing UV levels which also has implications for human health.*
Suggested correction has been taken into account in the revision of the Introduction.

*Page 5, Eq. 2 and 3: please specify the meaning of the different terms in these equations. Also, R is said to be the actual Sun-Earth distance, but R is also used as a subscript referring to Rayleigh airmass and optical depth. Maybe you could use 'r' for Rayleigh instead of 'R'?*
Explanation of terms in the equations has been improved. Rayleigh scattering terms are indicated by '*r*' in revised manuscript.

*Page 6, line 28: Where do the values for the Angstrom parameters come from?*
Justification for the used Ångström parameters has been added.

*Page 8, Eq. (5): (In revised manuscript Eq. 7.) should it not be $ln(V(\lambda)R^2)$ instead of $ln(V(\lambda))$ and $ln(V(0,\lambda))$ instead of $ln(R^2 V(0,\lambda))$?*
Suggested correction has been applied. Thanks!

*Page 8, Eq. (6): (In revised manuscript Eq. 8.) $p/p_0$ enters the Rayleigh scattering part. But then $\delta(R, \lambda)$ represents the Rayleigh scattering coefficient and not the Rayleigh optical depth.*
Good point. $p/p_0$ has been removed from the equation, and text has been adjusted.

*Page 9, line 7: Should this not be equation 6 instead of 5?*

Yes. Error corrected. (It is Eq. 8 in revised manuscript.)

*Page 9, Eq. (8): why is there no term for the NO2 and SO2 airmasses mN and mS? (for ozone, Rayleigh and aerosol, you specify an uncertainty for both the optical depth and the airmasses separately.) Is it because they are assumed to be very small in comparison to the other terms?*
Uncertainty terms for the $NO_2$ and $SO_2$ air masses have been added to Eq. 10 (former Eq. 8). As indicated by the Referee they are however considered small enough to be neglected. Text justifying this has been added in the solar position and air mass uncertainty section (3.4.6.).

*Page 10-11, section 4.2: (Section 3.4.2 in the revised manuscript.) For me it is not always clear how you obtain the actual values of the uncertainties of the contributing factors. For instance, for the first factor (the spread in the Langley Plots), you explain that you assume a triangular distribution for the V0s and take values close to the max and min of the individual V0s as limits and then determine the uncertainty which is [2.2 1.3 1.7 1.1]/$6^{1/2}$ %. How do you determine/chose the values used as 'close to the max and min'? Also, if possible, it would be nice if you could clarify this with a figure. Could you maybe clarify the entire paragraph a bit more to make it more understandable for readers who are not so familiar with uncertainty analysis calculations and the statistics behind it?*
A figure of $V_0$ distribution has been added. Attempt has been made to clarify the text regarding the estimation of $V_0$ uncertainty (old Sect 4.2, new Sect 3.4.2).

Page 11, section 4.3: The suggestion to include a formula for the calculation of Rayleigh optical depth uncertainty has been adopted.

*Page 13, for ozone airmass, you take into account the contribution of assuming an incorrect effective ozone altitude. If I understand correctly, you did not take this (an incorrect altitude for Rayleigh) into account for the uncertainty calculation of Rayleigh airmass. Why not? Is it included in the uncertainty due to algorithm uncertainty?*
It is a good point that a discussion about the effect of different atmospheric conditions on $m_r$ (relative optical air mass for Rayleigh scattering) was missing. It was already included in $u(m_r)$. Now also explaining text has been added.

Page 14, line 24: Justification for the use of the chosen values on α and β have been added to the text.

*Table 2: are the values between brackets not the standard error of the mean V0 (instead of the standard deviation)?*
The notation recommended by the GUM (2008) is followed here.
http://www.bipm.org/en/publications/guides/gum.html
In annex B, section B.2.17 "experimental standard deviation" and "experimental standard deviation of the mean" are defined. In Note 3, the notation "standard error of the mean" is stated to be incorrect. As far as we have understood, we are referring to the "experimental standard deviation of the mean", as noted in Sect. 2.2 (Sect. 3.1 in the revised manuscript).

Technical corrections

Page 1, line 11: "UVPFR" explanation added.

Page 1, line 14: Suggested correction applied.

Page 1, line 22: Suggested correction applied.

*Page 1, line 28: The term UV is already used in the abstract, I would move the explanation 'ultraviolet (UV)' to the abstract.*
Explanation of "UV" moved to abstract.

Page 2, line 6: The first paragraphs of the Introduction have been rewritten and the comment does not apply anymore.

Page 2, line 8-10: Suggested correction applied.

Page 2, line 11: Suggested correction applied.

Page 2, line 14: Suggested correction applied.

Page 2, line 17: Suggested correction applied.

 Page 2, line 17 (2): Suggested correction applied.

Page 2, line 19: Suggested correction applied.

Page 2, line 24: Suggested correction applied.

Page 2, line 26: Suggested correction applied.

Page 2, line 26(2): Suggested correction applied.

Page 2, line 27: Suggested correction applied.

Page 2, line 27(2): Suggested correction applied.

Page 3, line 4: Explanation of "PMOD/WRC" has been moved to where it is first used.

Page 3, lines 12-13: Suggested correction applied.

Page 3, line 17: Suggested correction applied.

Page 4, lines 2-5: Suggested corrections applied.

Page 4, line 11: Abbreviation explained.

Page 4, line 15: Suggested correction applied.

Page 4, line 20: Suggested correction applied (combined with suggestion from Referee #2)

Page 5, line 1: Suggested correction applied.

Page 5, line 1(2): Sentence rewritten, only in a slightly different way than suggested.

Page 5, line 2: Suggested correction applied.

Page 5, line 10: Suggested correction applied.

Page 5, line 16: Suggested correction applied.

*Page 6, line 5: the percentage for the longest wavelength (0.44%) is not in agreement with the value in table 2 (0.42%).*
0.44 % is the correct value, Table 2 has been corrected.

Page 6, line 10: Suggested correction applied.

Page 6, line 17: "VIS" and "NIR" are now specified in the Introduction.

Page 6, line 17(2): Suggested correction applied.

Page 7, line 3: Suggested correction applied.

Page 7, line 4: Suggested correction applied.

Page 8, lines 4-6: Suggested correction applied.

Page 8, line 28: Suggested correction applied.

Page 9, line 6: Suggested correction applied.

Page 10, line 27: Suggested correction applied.

Page 10, lines 28-30: Suggested correction applied.

*Page 11, line 11: Should there be '%' after the values?*
Yes, "%" is added.

Page 12, line 7: Suggested correction applied.

*Page 13, line 4: Reference Zerefos et al. 2016 is not yet in your list of references. And I guess you referred to the discussion paper? You can change this to Zerefos et al. 2017 as the paper is now officially published in ACP.*
Zerefos et al. 2017 is now referred to in the text and is now listed in the References.

Page 13, lines 10-11: Suggested correction partly applied.

Page 15, line 5: Suggested correction applied.

Page 15, lines 26-27: Suggested correction applied.

Page 16, line 4: Phrasing changed to "To be more specific, …".

Page 16, lines 13-15: Suggested correction applied.

Page 17, line 2: Suggested correction applied.

Page 17, line 3: Suggested correction applied.

Page 17, line 12-14: Suggested correction applied.

Page 17, line 14: Suggested correction applied.

Page 17, line 24: Suggested correction applied.

Page 18, line 3: Suggested correction applied.

Table 1 – Caption: Suggested correction applied.

Figure 1 – Caption: Suggested correction applied.

Figure 3 – Caption: Suggested correction applied.